
# Low-level liquid cloud properties during ORACLES retrieved using airborne polarimetric measurements and a neural network algorithm

Daniel J. Miller[1], Michal Segal-Rozenhaimer[2,3,4], Kirk Knobelspiesse[1], Jens Redemann[5], Brian Cairns[6], Mikhail Alexandrov[6,7], Bastiaan van Diedenhoven[6,8], and Andrzej Wasilewski[6,9]

[1]NASA Goddard Space Flight Center, Greenbelt, MD, USA
[2]NASA Ames Research Center, Moffett Field, CA, USA
[3]Bay Area Environmental Research Institute, Moffett Field, CA, USA
[4]Geophysics Dept., Porter School for the Environment and Earth Sciences, Tel-Aviv University, Tel-Aviv, Israel
[5]School of Meteorology, The University of Oklahoma, Norman, OK, USA
[6]NASA Goddard Institute for Space Studies, New York, NY, USA
[7]Applied Physics and Applied Mathematics Dept., Columbia University, New York, NY, USA
[8]Center for Climate System Research, Columbia University, New York, NY, USA
[9]SciSpace, LLC, Broadway, New York, NY, USA

**Correspondence:** Daniel J. Miller (Daniel.J.Miller@NASA.gov)

**Abstract.** In this study we developed a neural network (NN) that can be used to relate a large dataset of multi-angular and multi-spectral polarimetric remote sensing observations to retrievals of cloud microphysical properties. This effort builds upon our previous work, which explored the sensitivity of neural network input, architecture, and other design requirements for this type of remote sensing problem. In particular this work introduces a framework for appropriately weighting total and

polarized reflectances, which have vastly different magnitudes and measurement uncertainties. The NN is trained using an artificial training set and applied to Research Scanning Polarimeter (RSP) data obtained during the ORACLES field campaign (Observations of Aerosols above Clouds and their Interactions). The polarimetric RSP observations are unique in that they observe the same cloud from a very large number of angles within a variety of spectral bands resulting in a large dataset that can be explored rapidly with a NN approach. The usefulness applying a NN to a dataset such as this one stems from the possibility

of rapidly obtaining a retrieval that could be subsequently applied as a first-guess for slower but more rigorous physical-based retrieval algorithms. This approach could be particularly advantageous for more complicated atmospheric retrievals –such as when an aerosol layer lies above clouds like in ORACLES. For the ORACLES 2016 dataset comparisons of the NN and standard parametric polarimetric (PP) cloud retrieval give reasonable results for droplet effective radius ($r_e$ : $R = 0.756$, RMSE $= 1.74\,\mu$m) and cloud optical thickness ($\tau$ : $R = 0.950$, RMSE $= 1.82$). This level of statistical agreement is shown to

be similar to comparisons between the two most well-established cloud retrievals, namely the the polarimetric cloud retrieval and the bispectral total reflectance cloud retrieval. The NN retrievals from the ORACLES 2017 dataset result in retrievals of $r_e$ ($R = 0.54$, RMSE $= 4.77\,\mu$m) and $\tau$ ($R = 0.785$, RMSE $= 5.61$) that behave much more poorly. In particular we found that our NN retrieval approach does not perform well for thin ($\tau < 3$), inhomogeneous, or broken clouds. We also found





that correction for above-cloud atmospheric absorption improved the NN retrievals moderately - but retrievals without this correction still behaved similarly to existing cloud retrievals with a slight systematic offset.

## 1  Introduction

Advancing the scientific understanding of aerosol-cloud interactions is imperative for forming a more complete picture of the Earth climate system. These interactions are responsible for large uncertainties in our understanding of anthropogenic climate forcing (IPCC, 2013). Where the uncertainty primarily stems from the semi-direct and indirect effects of aerosols on clouds (Wilcox, 2010, 2012; Lu et al., 2018), which have been found to have significant yet uncertain climate impacts(Sakaeda et al., 2011).

Not many other regions of the world have as consistent aerosol-cloud interactions as the marine boundary layer of the southeast (SE) Atlantic Ocean. This region is dominated by a semi-permanent subtropical stratocumulus (Sc) deck that regularly interacts with significant biomass burning (BB) aerosols originating from natural and anthropogenic (agricultural) fires in central Africa during austral spring (July-October) (Zuidema et al., 2016). The aerosols are lofted into the mid-troposphere over land before being transported by large-scale circulation, eventually arriving above the marine stratocumulus deck (Adebiyi

and Zuidema, 2016). This leads to near persistent above-cloud aerosol (ACA) conditions that have consequential impacts on the radiative budget via direct radiative effects (i.e., enhanced aerosol absorption (Meyer et al., 2013; Zhang et al., 2016)) and semi-direct radiative effects that can induce numerous cloud adjustments (e.g., increased vertical stability, burn off, etc. (Koch and Del Genio, 2010; Wilcox, 2012)). As a result of this unique environment, the SE Atlantic region has become the focus of sustained research efforts. In addition to orbital observations, several international field campaigns have overlapped with

one another to explore this region, including: CLARIFY (U.K. Met Office, Cloud-Aerosol-Radiation Interactions and Forcing; Zuidema et al. (2016)), AEROCLO-SA (French National Research Agency, Aerosol Radiation and Clouds in Southern Africa; Formenti et al. (2019)), ONFIRE (U.S. National Science Foundation, Observations of Fire's Impact on the Southeast Atlantic Region), LASIC (U.S. Department of Energy, Layered Atlantic Smoke Interactions with Clouds; Zuidema et al. (2018)), and ORACLES (NASA, Observations of Aerosols above Clouds and their Interactions; Zuidema et al. (2016)). The latter of these

campaigns is the focus of this study.

To study this region, numerous state-of-the-art *in situ* and remote sensing instruments have participated in ORACLES flights in three deployments each austral spring from 2016 to 2018. As a consequence, the ORACLES dataset offers the opportunity to test and develop new remote sensing techniques – opening up the possibility of extending regional understanding to future satellite missions capable of making observations over global spatial- and climactic time-scales. For example, the upcoming

NASA Plankton, Aerosol, Clouds and ocean Ecosystem (PACE) mission, which will deploy instruments with similar capabilities as the one we will focus on in this study. From a passive cloud remote sensing perspective, the persistence of ACA in



the ORACLES study region can represent a difficult and sometimes confounding issue. Cloud microphysical retrievals which do not consider the presence of the aerosol above the cloud can suffer biases due to the impact of absorption of the overlying BB aerosols in shortwave spectral bands. Most notably, this was found to be the case for Moderate Resolution Imaging Spectroradiometer (MODIS) cloud retrieval product (Meyer et al., 2013). It is possible to correct for this impact, but an assumed

aerosol model is required to constrain the otherwise unknown optical properties of the aerosol. On the other hand, there are some ACA retrieval methods that attempt to simultaneously retrieve full aerosol and cloud properties of ACA scenes. However, the existing techniques each still exhibit shortcomings when it comes to constraining aerosol absorption properties (e.g., single scattering albedo or complex refractive index) and thus can result in an inaccurate representation of the direct radiative effect of ACA (Knobelspiesse et al., 2015; Yu and Zhang, 2013). One of the more promising approaches takes advantage of the large in-

formation content of multi-spectral, multi-angular, and polarization observations. The vast information content of polarimetric observations provides ample opportunities to simultaneously retrieve aerosol and cloud properties. This methodology has been applied to both orbital (Waquet et al., 2009, 2013) and sub-orbital field campaign observations (Knobelspiesse et al., 2011b; Xu et al., 2018).

    In this study, we make use of polarimetric observations obtained using the Research Scanning Polarimeter (RSP) during

ORACLES 2016 and 2017 field campaigns. The RSP is the airborne prototype for the Aerosol Polarimetry Sensor (APS) built for the NASA Glory Mission (Mishchenko et al., 2007; Peralta et al., 2007; Persh et al., 2010). While Glory did not successfully enter orbit due to a launch failure, the pair of RSP instruments, denoted RSP1 and RSP2, continue to make observations and have been deployed on over 25 field missions in the last twenty years. The instruments heritage, accuracy, and measurement characteristics make it well suited for observations of clouds (Alexandrov et al., 2012a, b; van Diedenhoven et al., 2016;

Diedenhoven et al., 2013; Sinclair et al., 2017), aerosols (Chowdhary et al., 2001; Chowdhary and Cairns, 2002; Chowdhary et al., 2012; Knobelspiesse et al., 2011a, b; Wu et al., 2015, 2016; Stamnes et al., 2018), the ocean (Chowdhary et al., 2006, 2005b, a; Ottaviani et al., 2012) and snow (Ottaviani et al., 2015). In particular the cloud retrieval products of RSP are well established and validated (Alexandrov et al., 2015, 2016). In contrast, the retrieval of ACA properties has been implemented and tested only in a few case studies (Knobelspiesse et al., 2011b; Pistone et al., 2019).

The main limitation to the latter effort is the high computational expense, requiring numerous iterative calls to a time-consuming forward radiative transfer (RT) model. These iterative calls are made in an effort of match observations to a simulated scene, thereby retrieving optical and microphysical properties of the cloud and aerosol layers concurrently. Additionally, the dimensionality of the observational data (large for multi-angle polarimetry) as well as the number of variables that are retrieved (large for ACA retrieval) can significantly slow down this type of approach. As a consequence of these computational

limitations, accelerating these types of algorithms is critical to developing a useful retrieval product. Here, the NN retrieval approach is useful, since it offers some important benefits and can be complementary to the solutions discussed above. First, it can be used to explore the non-linear relationships between observation variables and retrieval properties in a manner unbiased by prior understanding of that relationship – providing unique insight to other inverse approaches. Second, after the network is trained, it is capable of transforming a vector of observed variables to retrievals rapidly by applying the "transfer function"

resulting from the trained network. Third, the NN retrieval can serve as a prior state vector for an optimal estimation retrieval,





accelerating and improving its results, as demonstrated by (Di Noia et al., 2015), for a NN retrieval of aerosol properties using a multi-angular and multi-wavelength polarized ground-based instrument.

Here, we are capitalizing on our previous work in Segal-Rozenhaimer et al. (2018), where we have developed a NN retrieval scheme for low-level cloud properties. By focusing on clouds only, we can easily compare our results to the other RSP cloud

retrieval algorithms to gain an understanding of how the NN retrieval is performing. Our original NN scheme was used twice, first as a base architecture for a sensitivity study, and second, as a retrieval scheme for low-level cloud properties during ORACLES 2016. The sensitivity study addressed numerous aspects in the algorithm design such as the type of input variables and their dimensionality, while the retrieval scheme used a preliminary (and somewhat limited) NN training set. Perhaps the most important outcome from this work was the determination of the type of input data required to use in a NN to retrieve cloud

properties. For example, it is not necessarily obvious what pair of independent polarimetric observations would work best for a NN approach. We were able to demonstrate that the NN trained retrievals with the lowest root mean square error (RMSE) and highest correlation were found with inputs of the total reflectance ($R_\mathrm{I}$) and Degree of Linear Polarization (DoLP)(Segal-Rozenhaimer et al., 2018). It is also worth emphasizing here that the existing passive cloud microphysical retrievals (e.g., bispectral (Nakajima and King, 1990) and polarimetric (Alexandrov et al., 2012a)), either utilize observations of total or

polarized reflectances separately to infer cloud droplet size distribution shape and cloud optical thickness. In contrast, this approach allows us to effectively mix the information contained in both total and polarized reflectance observations – resulting in a retrieval that attempts to be consistent for both observations. One major difference we are introducing in this work, compared to our previous NN, is the dimensionality of the input layer of the network. Previously, we used principal component analysis (PCA) to reduce the dimensionality of the input vector to improve the network in an attempt to increase convergence

and generalization capability, as suggested in many prior studies (Di Noia et al., 2015; Del Frate and Schiavon, 1999; Del Frate et al., 2005; LeCun et al., 1989). This was performed separately on the $R_\mathrm{I}$ and DoLP, which were then both used as an input to the NN. However, after training the network in this manner and applying it to a subset of ORACLES 2016 measurements, we found that the network placed more importance of $R_\mathrm{I}$ than on DoLP measurements, despite the fact that the uncertainty of the latter is much lower (0.2%) than the former (3%). This resulted in poor accuracy and highly biased retrievals of cloud droplet

size. In this work, we implemented a new approach to the network architecture that allows us to directly input the observation vector into the network – eliminating the need for dimensionality reduction and allowing us to treat disparate observational uncertainties in a more explicit manner.

The rest of the paper is organized in the following manner. Section 2 outlines the properties of the RSP instrument observations and uncertainties (section 2.1) as well as specifics regarding the data obtained during the 2016 and 2017 ORACLES

field campaigns (section 2.2). Additionally, in this section we also give an overview of the various standard cloud property retrieval products from the RSP instrument, which we use to compare with our NN based retrievals (section 2.3). Section 3 focuses on new developments and improvements implemented in our approach to the NN retrieval scheme. Section 4 focuses on the output of the NN and the comparison of the NN retrievals to RSP's existing cloud retrievals during ORACLES 2016 (section 4.2 and ORACLES 2017 (section 4.3). Finally, in section 5 and 6 we summarize our findings and discuss strengths,

limitations, and indicate future goals of this research.



## 2 Data and Methods

### 2.1 Research Scanning Polarimeter

The Research Scanning Polarimeter is an airborne multi-angular polarimetric instrument that continuously scans in the along-track direction, resulting in 152 views of each pixel at viewing zenith angles (VZA) up to $\pm 60°$ (forward and aft of the flight direction). As a result, the RSP instrument has a very high angular resolution of $\Delta\theta = 0.802°$. Measurements of the total and polarized reflectances are obtained at nine visible and SWIR spectral channels with the following band centers: $0.410, 0.470, 0.555, 0.670, 0.865, 0.960, 1.59, 1.88, 2.26, \mu m.$[1]

Observed reflectances are defined in terms of the Stokes vector elements describing linearly polarized light (I, Q, and U) and are unitless due to normalization with respect to the incident solar irradiance in the following manner:

$$R_I = I \frac{\pi r_0^2}{F_0 \cos(\theta_0)}, \tag{1}$$

$$R_Q = Q \frac{\pi r_0^2}{F_0 \cos(\theta_0)}, \tag{2}$$

$$R_U = U \frac{\pi r_0^2}{F_0 \cos(\theta_0)}. \tag{3}$$

Where $R_\mathrm{I}$ is the total reflectance (including unpolarized and polarized light) and $R_\mathrm{Q}$ and $R_\mathrm{U}$ are the two perpendicular components of the linearly polarized reflectance. Additionally, $r_0$ is the earth-sun distance in Astronomical Units, $F_0$ is the top of atmosphere solar irradiance, and $\theta_0$ is the solar zenith angle (SZA). It is important to note that the magnitudes of the linearly polarized reflectances ($R_\mathrm{Q}$, $R_\mathrm{U}$) are initially defined in an instrument polarization reference frame. In this work we transform from the instrument reference plane to the principal scattering plane (hereafter simply the principal plane), which is the plane containing both incident solar and and observation viewing direction vectors. In the principal plane the single-scattered polarized reflectance of cloud droplets is fully described by $R_\mathrm{Q}$ with measurements of $R_\mathrm{U}$ expected to be near zero in magnitude (Mishchenko et al., 2007). However, for observations off of the principal plane the polarized reflectance is distributed between both $R_\mathrm{Q}$ and $R_\mathrm{U}$. One way to separate the dependence on a geometric reference is to decompose the polarized reflectance measurements into the magnitude (independent of reference) and angle of the polarization vector (dependent on reference). For our purposes, the angle of the polarization vector is not particularly important, and the magnitude of the linearly polarized reflectance ($R_\mathrm{P}$) is the measurement of interest,

$$R_\mathrm{P} = \sqrt{R_\mathrm{Q}^2 + R_\mathrm{U}^2}. \tag{4}$$

---

[1]For the purposes of this study we will neglect the $0.960$ and $1.88\,\mu m$ bands as they are primarily used for the retrieval of column water vapor concentrations



Additionally, it is also convenient to introduce the degree of linear polarization (DoLP), which is the ratio of the magnitude of polarized reflectance to the total reflectance,

$$\mathrm{DoLP} = \frac{R_\mathrm{P}}{R_\mathrm{I}}. \tag{5}$$

The uncertainties in $R_\mathrm{I}$ and DoLP differ from one another by an order of magnitude, with $\delta R_I \approx 3\%$ and $\delta\mathrm{DoLP} \approx 0.2\%$
respectively. For $R_\mathrm{I}$ this measurement uncertainty is largely a result of radiometric calibration uncertainty. Whereas because DoLP is a relative measurement, calibration uncertainty is less important and sensitivity to random noise becomes the dominant source of uncertainty. A more complete description of RSP uncertainty and uncertainty models for the instruments can be found in Knobelspiesse et al. (2019).

As mentioned previously in section 1, RSP2 flew throughout the ORACLES mission. In 2016, RSP2 was on board the NASA
ER2, but in 2017 and 2018 it was moved to the NASA P3. It is worth noting that RSP1 was also deployed during the ORACLES 2016 campaign on board the P3, however there were data collection problems. Unexpected wind resistance at the instrument scanning assembly prevented it from spinning at the required rate, resulting in incomplete scans and poor geo-registration. Successful data collection occurred for a small portion of the flights, but the limited nature of these observations did not justify application of the NN. RSP2 on the ER2 had no significant issues throughout the 2016 campaign.

In practice, RSP is not oriented in the aircraft such that there is a symmetric range of VZA about nadir viewing (i.e., 152 measurements spanning $\pm60°$). Rather, due to mounting constraints that result in aircraft vignetting, it is often positioned such that the range is $[+50° : -70°]$ (forward to aft). For this reason, we restrict ourselves to a reduced range of angles that are symmetric about nadir (112 measurements spanning $\pm45°$). This restriction is important for our application, as it makes it possible to use the same NN for any heading.

RSP reflectances used in the NN dataset are aggregated to cloud top height following the same procedure as described in fig. 1 of Alexandrov et al. (2012a).

## 2.2 Data from the ORACLES Deployments

The datasets obtained during the first two years of ORACLES deployments in 2016 and 2017 differ from each other substantially. By design, each deployment of the ORACLES campaign was intended to target and characterize different months
during the BB season (July through October), where the prevailing easterly wind transports the BB aerosols from sub-Saharan Africa fire events to the SE Atlantic, where the stratocumulus cloud deck is located (Swap, 1996; Costantino and Breon, 2013; Painemal et al., 2014; Zhang et al., 2016; Zuidema et al., 2016). To that end, the peak of the season (September) was the focus of the 2016 deployment, while the beginning of the season (August) was the focus of 2017, and finally the end of the season (October) is the focus of the 2018 deployment. Additionally, from a logistical perspective, flight operations were not based out
of the same location during each deployment. In 2016, flight operations were based out of Walvis Bay, Namibia (figure 2.2, dotted lines) and in 2017 (and also 2018) flight operations were moved north of the study region to the island of São Tomé (figure 2.2, dashed lines). The consequence of this logistical change is that there are regional differences in the cloud proper-



ties observed throughout the campaign. Walvis Bay is located close to climatological center of the stratocumulus deck during the biomass burning season. Whereas flights out of São Tomé in 2017 (and also in 2018) typically had to fly further south before encountering the stratocumulus cloud deck. As a consequence, from an environmental perspective, the clouds observed during the ORACLES 2016 field campaign were largely overcast marine stratocumulus but flights during 2017 observed less

homogeneous marine boundary layer clouds associated with the transition between stratocumulus and broken cumulus cloud regimes. In addition to the regional differences, the behavior in the SE Atlantic changes to a greater extent seasonally and to a lesser extent interannually. Seasonally, the stratocumulus deck in this region shifts southward later in the season with the cloud fraction maximum occurring in September (Wood, 2012). In an interannual sense, the stratocumulus deck is modified by changes in lower tropospheric stability (LTS) that can be strongly correlated with sea surface temperature and free tropospheric

temperature (Wood, 2012). Because the ORACLES campaign spanned multiple years and different seasons, the role of inter-annual variability is important to consider. However, for the purposes of this study, all of variabilities result in greater diversity in the cloud retrieval dataset, which we can use to gain a better understanding of the behavior of our retrieval approach under a variety of cloudy conditions.

From an instrument perspective, the RSP flew on board different flight platforms during the 2016 and 2017 deployments. In

2016, the NASA high-altitude ER2 was dedicated to remote sensing instruments, obtaining data from a near consistent flight altitude above 18 km. On the other hand, during 2017 (and 2018) the RSP flew on board the NASA P3 at a more variable range of altitudes because the P3 sampled throughout the boundary layer, in the cloud, in the aerosol layer, and above the cloud. As a consequence, the NN training sets for these two years differ from one another in order to be appropriately tailored to the airborne platform differences due mainly to their different altitudes and the Rayleigh scattering differences. The training set

for ORACLES 2016 was created for a constant aircraft altitude of 20 km whereas the training set for ORACLES 2017 was constructed to account for level legs at different aircraft altitudes. The differences in the training set definition for each of these two datasets is further discussed in section 3.1. Note that while ORACLES 2018 data is now available it had not been available until after the analysis of the this NN implementation was complete. However, the 2018 NN results will also be available in our data archive when they are completed (refer to the data availability section for a link to the data archive).

As with any field campaign, instrument-specific complications arose that need to be considered. For example, the SWIR detectors of RSP must be cooled to obtain SWIR reflectances without significant noise, however, during the 2017 field campaign there was a lack of liquid nitrogen to cool the detectors during some of the flights. As a consequence, much of the 2017 dataset lacks data from the SWIR channels. To explore the consequence of the loss of the SWIR channels on our retrievals we created two different training datasets for our ORACLES 2017 NN retrieval scheme; one excluding the SWIR channels (applied on the

entire dataset), and one that included the SWIR channels during training (applied on the flights that acquired data with these channels).

Before performing the comparison of different retrieval methods, presented in section 4, RSP data is first screened for a number of conditions to obtain useful comparable retrieval data-sets. The philosophy behind this screening process is to obtain the best data for usage in this study but at the same time not cast aside NN retrievals that may be useful in future studies. In

addition to RSP data, we also use cloud top height data from the NASA Langley airborne second-generation High Spectral





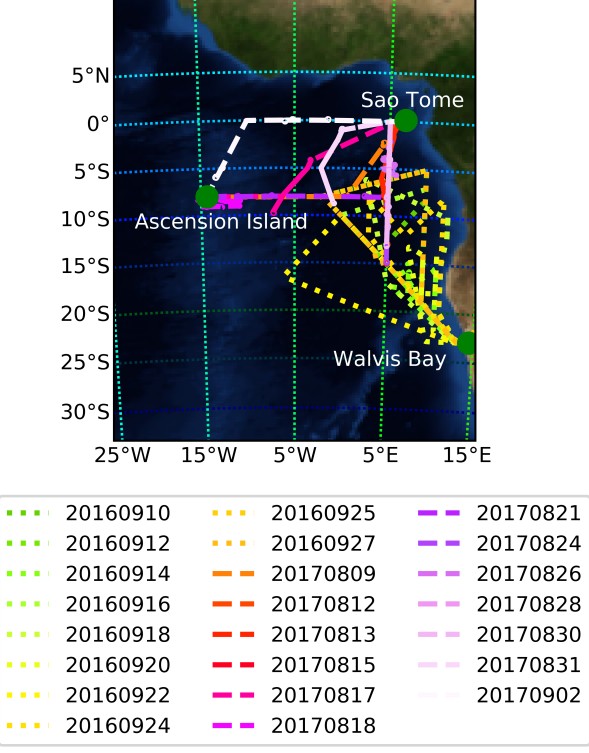

**Figure 1.** Flight tracks and study regions for the ORACLES 2016 (dotted lines) and 2017 (dashed lines) field campaigns. Additionally, key take-off and landing locations are indicated and labeled with green circles. Map data based on the Blue Marble: Next Generation from the NASA Earth Observatory.

Resolution Lidar (HSRL-2) to remove observations of high-level or multi-layer clouds in an attempt to limit the retrieval to low-level marine boundary layer clouds (Hair et al., 2008; Burton et al., 2018). To that end, the following screening criteria are applied to the datasets compared:

– Cloudy scenes as identified by other RSP retrieval methods

5    – Successful RSP retrievals using all other techniques

– Coincident RSP and HSRL-2 data for cloud top height definition

– Instances with HSRL-2 cloud top height below 2 km

In a few limited cases coincident HSRL-2 and RSP data was not available, which precludes some retrieval data from the screening criteria above. This screening criteria was removed in the final data product (refer to the data availablility section)
10   so that it could include NN retrievals for the entire RSP dataset. Also included with the dataset is a guide discussing how to evaluate the screening flags and select data suitable for other uses.



### 2.3 Standard RSP Cloud Retrievals

The shortwave radiative impact of clouds largely depends on microphysical-scale cloud properties that define the droplet size distribution (DSD) (Twomey, 1977). Additionally, the DSD also plays an important role in cloud-precipitation processes (Pruppacher and Klett, 1978). In cloud remote sensing it is common to describe the cloud droplet size distribution using the

gamma distribution presented in Hansen and Travis (1974), because it is both mathematically convenient and fits well to *in situ* observations (Deirmendjian, 1964; Tampieri and Tomasi, 1976),

$$N(r) = N_0 C r^{(1-3v_\mathrm{e})/v_\mathrm{e}} \exp\left[-\frac{r}{r_\mathrm{e} v_\mathrm{e}}\right], \tag{6}$$

$$C \equiv \left((r_\mathrm{e} v_\mathrm{e})^{(1-2v_\mathrm{e})/v_\mathrm{e}} \Gamma\left[(1-2v_\mathrm{e})/v_\mathrm{e}\right]\right)^{-1}. \tag{7}$$

This is a three parameter distribution characterized by a droplet number concentration ($N_0$, $[1/\mathrm{cm}^3]$), a droplet effective radius

($r_\mathrm{e}$, $[\mu\mathrm{m}]$), and a droplet effective variance ($v_\mathrm{e}$, [-]). The normalization constant for this distribution, $C$, is calculated based on these parameters and the gamma-function ($\Gamma$). The effective radius, is a cross-section weighted droplet size that, for the purposes of light scattering applications is usefully related to the scattering droplet size described in Hansen and Travis (1974). The effective variance is related to the droplet size distribution dispersion and can also be interpreted as a measure of the asymmetry of the droplet size distribution.

$$r_\mathrm{e} = \frac{\langle r^3 \rangle}{\langle r^2 \rangle}, \tag{8}$$

$$v_\mathrm{e} = \frac{1}{r_\mathrm{e}^2} \frac{\langle (r - r_\mathrm{e})^2 r^2 \rangle}{\langle r^2 \rangle}. \tag{9}$$

The existing RSP liquid cloud retrieval products include three very different methods of inferring cloud microphysical information. Each of these methods differ from one another in fundamental ways that include: integrating observational data of different types (i.e., total or polarized reflected light), capability of retrieving different combinations of variables (i.e., some

combination of $r_\mathrm{e}$, $v_\mathrm{e}$, and $\tau$), and sensitivities (i.e., to cloud vertical profile, aerosol above cloud or microphysical regime).

The first method, often referred to as the bispectral Nakajima-King (NJK) method, is an approach that takes advantage of a difference in sensitivity to cloud optical thickness and effective radius in a pair of spectral total reflectance bands (Nakajima and King, 1990). One band is in a scattering-dominated visible to near infrared (VNIR) band while the other is in a more absorptive shortwave infrared (SWIR) band. The NJK retrieval performed by RSP makes use of nadir viewing total reflectances

in $0.865\,\mu\mathrm{m}$ and the $2.26\,\mu\mathrm{m}$ or $1.59\,\mu\mathrm{m}$ spectral bands. For the purposes of this study, we focus on the RSP NJK retrieval using the in $0.865\,\mu\mathrm{m}$ and the $2.26\,\mu\mathrm{m}$ spectral band combination. This retrieval, most notably implemented for the MODIS cloud retrieval product, is typically performed as a two-dimensional interpolation of observed reflectances within a discrete look up table (LUT), relating reflectances to unique pairs of $r_\mathrm{e}$ and $\tau$ values (Platnick et al., 2016). This particular method is also important because it obtains a retrieval of cloud optical thickness, while the following two other methods, which are





based on polarized reflectances, retrieve only droplet size distribution information ($r_e$, and $v_e$). As a consequence, these other methods secondarily perform an optical thickness retrieval in a manner similar to the NJK retrieval but with a single VNIR band LUT with a pre-constrained $r_e$ obtained via a polarimetric retrieval. In the context of the ORACLES field campaign it is also important to emphasize that the NJK method has been shown to be systematically biased by the presence of ACA – resulting in a high bias in both $r_e$ and $\tau$ retrievals that is highly dependent on aerosol model assumptions, especially those that can impact absorption (e.g., aerosol single scattering albedo or refractive index) (Meyer et al., 2013).

The second RSP retrieval, referred to here as the parametric polarimetric method (PP), makes use of a library of calculations that describe the angular distribution of single-scattered polarized light (known as polarized phase functions, $-P_{12}$). The phase functions and reflectances are both characterized by angular rainbow features (appearing between scattering angles of $130°$ and $170°$) that predictably shift and erode depending on the properties of the particular droplet size distribution (i.e, the $r_e$ and $v_e$ pair) (Bréon and Goloub, 1998). Because polarized reflectances are dominated by single scattering, this library can be used to obtain a best fit solution that matches the observed multi-angular $R_P$ or DoLP in a single spectral band. The phase function is then modified by parametric functions that account for Rayleigh scattering and multiple scattering effects. The best fit solution of this parametric phase function to the observed multi-angular reflectance corresponds to the droplet size distribution parameters retrieved (Alexandrov et al., 2012a). The PP retrieval can be performed for a number of different spectral bands, however, in this study we make use of the retrieval performed for the 0.865 $\mu$m band. This is because the longer shortwave spectral bands have been shown to be more sensitive to a greater range of droplet sizes at a fixed angular resolution (Miller et al., 2018).

The third RSP retrieval method is a non-parametric approach, known as the Rainbow Fourier Transform (RFT), that retrieves the droplet size distribution in a functional form via a mathematical transformation mapping the polarized reflectance in angular space to the droplet size distribution in microphysical space. As the name indicates, this approach is similar to the relationship between oscillatory signals (frequency space) and their corresponding Fourier transforms (amplitude space) (Alexandrov et al., 2012b). This method is useful for evaluating the assumption that droplet size distributions are well-behaved and mono-modal – an implicit assumption for both of the gamma-distribution parameter retrievals discussed previously (Alexandrov et al., 2016). The RFT retrieval reports the distribution shape, but it also reports the best fit gamma-distribution parameters of the two most prominent modes of the size distribution, resulting in $r_e$ and $v_e$ retrievals for each mode. When we discuss the RFT retrieval in this study as a single $r_e$ or $v_e$ value we are always referring to the most prominent mode of the size distribution.

The physical differences between NJK and PP cloud property retrievals was recently the topic of research in Miller et al. (2018). One of the findings of that study was that high spatial resolution retrievals ($50\,\text{m}$) mostly agreed with one another to within the measurement uncertainties of the two methods. However, at coarse spatial resolutions ($> 300\,\text{m}$) observations of spatially inhomogeneous cloud fields caused the NJK retrieval to be high-biased resulting in differences between the two retrieval approaches. In the context of this study, airborne observations made by RSP have quite a high spatial resolution (on the order of tens of meters to hundreds of meters depending on aircraft altitude), which should avoid some spatial inhomogeneity issues in this comparison. Another finding of Miller et al. (2018) was that there can be significant high biases for the NJK retrieval when droplet sizes become small ($r_e \approx 5\,\mu\text{m}$) or for optically thin clouds ($\tau < 3$). Given the high spatial and angular





resolution of the RSP retrievals in this study, it is likely that biases associated with the "small and thin" population will be the most prevalent source of bias in our data.

In this study we intend to make informed comparisons between these already existing retrievals and the NN retrieval. However, before doing that it is important to evaluate how these disparate retrieval products compare to one another. To

that end, figure 2 evaluates each of the retrievals against one another in much the same manner as in Miller et al. (2018). All of these comparisons are made using ORACLES data that has been previously screened for multi-layer clouds, as detailed in section 2.2. The comparison of NJK and PP retrievals of $r_e$ are shown as density regressions for the ORACLES 2016 (figure 2a) and ORACLES 2017 (figure 2d) datasets. From the ORACLES 2016 comparison it is evident that the two retrievals are similar to one another —with a correlation of $R = 0.747$, a mean bias of $-0.830 \, \mu m$, and a $\mathrm{RMSE} = 1.74 \, \mu m$. It is noteworthy that

despite being similar overall, the RMSE of the retrieval comparison is actually still quite large with figure 2b indicating that the $r_e$ retrieval bias is being driven by retrievals of the low $\tau$ population ($\tau < 3$). With that in mind, the statistics for comparisons of the two retrievals excluding the low $\tau$ population are significantly improved. The comparison for ORACLES 2017 is more complicated, with increased relative occurrence of thin clouds and increased spatial inhomogeneity the statistical metrics are much poorer – with a correlation of $R = 0.201$, a mean bias of $-1.41 \, \mu m$, and a $\mathrm{RMSE} = 3.38 \, \mu m$. However, this behavior is

still, to a large extent, associated with the low $\tau$ population with statistics improving when that population is excluded (looking only at $\tau > 3$) as indicated in figure 2. For both ORACLES 2016 (figure 2c) and ORACLES 2017 (figure 2f) datasets the comparison of $\tau$ reveals that there is typically very little relative bias. In some cases, there are biases observed between the two retrievals corresponding to small NJK $r_e$ retrievals – indicating that using the PP constrained $r_e$ retrieval produced a different $\tau$ retrieval. Given the statistical properties of the comparisons of these two well-established retrievals approaches we should

expect to be satisfied if we find a similar degree of agreement between the NN retrieval and any of the standard RSP retrievals.

## 3 Neural Network Development

As discussed in section 1, the NN architecture implemented in this study has changed significantly in response to the findings of our previous work (Segal-Rozenhaimer et al., 2018). In section 3.1 we will discuss the definition of the training set and particularities to the first two years of the ORACLES field campaign. Then, section 3.2 discusses our new approach to

preprocessing input observations and uncertainties of total and polarized reflectances. Finally, in section 3.3 we outline the architectural variables such as network structure, learning rate, etc.

### 3.1 Training Set Simulations

The synthetic observational data set used to train the NN is created using a vectorized radiative transfer (RT) model to generate total and polarized reflectances that mimic the conditions of the observations made by the RSP instrument during the

ORACLES field campaign. The RT model used in this study is the plane-parallel (1-D) polarized doubling-adding (PDA) model developed at the NASA Goddard Institute for Space Studies. This model is built upon the methods described in van de Hulst and Irvine (1963) and can efficiently solve radiative transfer problems in optically thick atmospheres (Hovenier; Hansen,



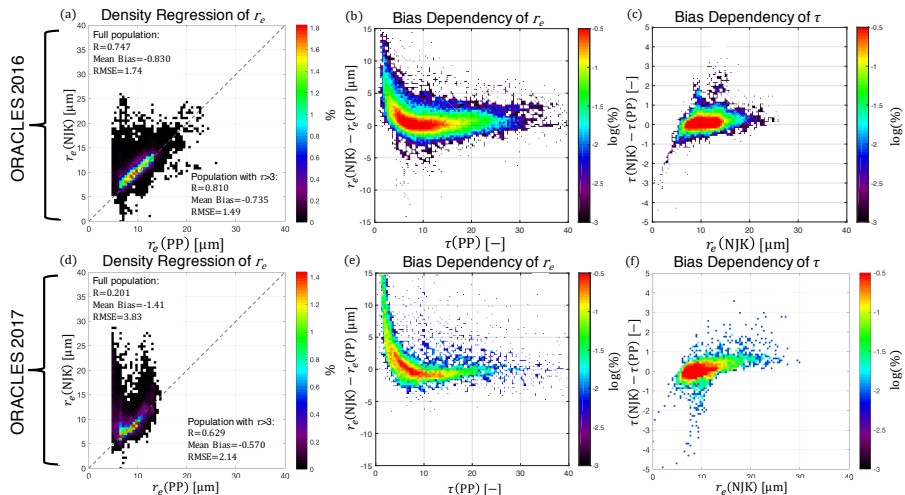

**Figure 2.** A series of comparisons between the PP (using the $0.865\,\mu m\ polarized\ reflectances$) and NJK retrievals (using the $2.26\,\mu m$ SWIR band) made by RSP during ORACLES 2016 (upper row) and 2017 (lower row). In the left two panels NJK $r_{\mathrm{e}}$ (y-axis) and PP $r_{\mathrm{e}}$ (x-axis) retrievals are compared using a density regression plot with color bar that indicates the percentage of observations contained in each bin and a dashed one-to-one line. In the top left these plots the correlation, mean bias, and RMSE are reported for the full retrieval population while the same statistics are also reported for thick clouds ($\tau > 3$) only in the bottom right. The rest of the panels use a different color bar that emphasizes features of smaller populations using the logarithm of the percentage of observations in each bin. In the middle two panels we display the bias between NJK and PP retrieval of $r_{\mathrm{e}}$ (y-axis) is shown with respect to the PP retrieval of $\tau$ (x-axis). Finally, in the right two panels the bias between NJK and PP retrievals of $\tau$ (y-axis) is shown with respect to the NJK $r_{\mathrm{e}}$ retrieval.

1971; Hansen and Travis, 1974; De Haan et al., 1987). This PDA radiative transfer code was also selected to be used for inversions during the Glory mission and therefore was specifically improved and tailored for polarimetric accuracy (Cairns and Chowdhary, 2003). As a consequence, it is very efficient at generating the simulated multi-spectral, multi-angular polarimetric observations required to mimic the observations of the RSP instrument.

5    The training sets for the operational NN were generated based on the range of cloud properties observed in ORACLES 2016 (from RSP and in-situ cloud measurements), and were tailored for each of the airborne platforms, as discussed in section 2.2. Compared to our training set generated in the Segal-Rozenhaimer et al. (2018) study, these training sets expand the relative azimuth angle (RAA) range significantly (from $[0 : 10°]$ to $[0 : 90°]$) as shown in table 1. This new RAA range covers all possible azimuth geometries, since radiative transfer is symmetric about the solar plane and because the RSP scans in both

10    forward and aft directions. For all training sets, cloud top height was fixed at 1 km, which was found to be a reasonable assumption based on other independent measurements during the ORACLES campaigns. Also, since the ER-2 is a high-altitude platform that flies at a constant altitude, the training set simulations (table 1) were made for a constant aircraft altitude of 20 km. However, since the P-3 is a low-altitude flying platform, altitude variations were much larger than the ER-2, and the training set was constructed to predict measurements obtained along a constant level legs of various altitudes (table 2). Additionally,





**Table 1.** Parameter grid space used to generate the training set ($N = 44,064$ cases) for the operational NN used for cloud retrievals from ER-2 during ORACLES 2016 field campaign. Aircraft altitude is set as constant at 20 km.

| Parameter [units] | # of grid points | Training Grid |
|---|---|---|
| $r_e$ [$\mu$m] | 6 | 5, 7.5, 10, 12.5, 15, 20 |
| $v_e$ [-] | 6 | 0.01, 0.03, 0.05, 0.07, 0.1, 0.15 |
| $\tau$ [-] | 6 | 2.5, 5, 10, 15, 20, 30 |
| SZA [°] | 12 | 10, 15, 20, 25, 30, 35, 40, 45, 50, 55, 60, 65 |
| RAA [-] | 17 | 0, 2, 4, 6, 8, 12, 16, 20, 24, 28, 32, 40, 50, 60, 70, 80, 90 |

**Table 2.** Parameter grid space used to generate the training set ($N = 261,144$ cases) for the operational NN used for cloud retrievals from P-3 during ORACLES 2017 field campaign.

| Parameter [units] | # of grid points | Training Grid |
|---|---|---|
| Aircraft Altitude [$m$] | 3 | 5000, 6000, 7000 |
| $r_e$ [$\mu$m] | 6 | 5, 7.5, 10, 12.5, 15, 20 |
| $v_e$ [-] | 6 | 0.01, 0.03, 0.05, 0.07, 0.1, 0.15 |
| $\tau$ [-] | 6 | 2.5, 5, 10, 15, 20, 30 |
| SZA [°] | 13 | 5 to 65 in increments of 5 |
| RAA [°] | 31 | 0 to 90 in increments of 3 |

there was more variability in cloud top height during 2017 as the clouds observed were often transitioning between low-level stratocumulus regime and into mid-level cloud regimes. Since the atmospheric scattering between the flying platform and the cloud top has an effect on the measured signals, the generated cases might not be optimal for all the scenes flown during 2017. The role that all of these training set decisions play in the behavior of our retrieval results will be discussed in section 4.

## 3.2 Pre-processing Input Observations

In our former NN retrieval scheme, we reduced the dimensionality of the input layer by reducing measurement vector inputs to principle components (PC) before introducing them as input to the NN (Segal-Rozenhaimer et al., 2018). Our improved retrieval scheme described here is instead trained with and applied to the measurement vector itself. This solution was conceived to allow more appropriate weighting of $R_I$ or DoLP, which have significantly different measurement uncertainties. The size of the input layer changed from 122 inputs (100 PC for DoLP, 20 for $R_I$, and the two geometry inputs, i.e. SZA and RAA for each case) to 1570 (concatenating $R_I$ and DoLP, each spanning 784 values, covering the 112 instrument viewing angles in seven wavelengths plus the two geometry input values). To accommodate this ten-fold increase in the size of the input layer, we implemented a new approach to our NN architecture, which will be discussed in section 3.3. The advantage of this approach is that it allows us to adequately scale (weight) the different input sources ($R_I$ and DoLP) according to their measurement uncertainty.





This is specifically important for polarimetric observations because both the magnitude and uncertainty of RSP observations of $R_I$ and DoLP differ by an order of magnitude. The uncertainty in $R_I$ is $\delta R_I \approx 3\%$ and is largely dominated by systematic calibration-dependent, biases; whereas the uncertainty in DoLP is $\delta \text{DoLP} \approx 0.2\%$ and is largely dominated by random noise that varies with scene reflectance ($R_I$). Without consideration of relative magnitude and uncertainty, a NN incorporating both

of these types of observations would erroneously rely too much on high magnitude and uncertainty $R_I$ observations at the expense of low magnitude and uncertainty DoLP. To avoid this issue, we incorporate knowledge of measurement uncertainty into the vector standardization that is typically used to pre-process data prior to NN training and application. Typically, inputs to the NN are standardized in the following manner,

$$\hat{x_i} = \frac{x_i - \bar{x}}{s}, \tag{10}$$

where $\bar{x}$ is the mean of all $x$ over elements $i$, and $s$ is the standard deviation. In contrast to this, we have modified this process so that measurement uncertainty is incorporated into the standardized data, such that the standard deviation is replaced by the expected measurement uncertainty of the mean observation obtained for same geometry and band $\bar{x}(\theta_0, \theta, \Delta\phi, \lambda)$,

$$\hat{x_i}(\theta_0, \theta, \Delta\phi, \lambda) = \frac{x_i(\theta_0, \theta, \Delta\phi, \lambda) - \bar{x}(\theta_0, \theta, \Delta\phi, \lambda)}{\sigma(\bar{x}(\theta_0, \theta, \Delta\phi, \lambda))}, \tag{11}$$

where the measurement uncertainty, $\sigma$, is calculated using the RSP uncertainty model in (Knobelspiesse et al., 2019). We have

explicitly noted the dimensions over which the average is calculated (solar zenith angle, view zenith angle, relative solar-view azimuth angle, spectral band, $(\theta_0, \theta, \Delta\phi, \lambda)$), such that a new standardization is calculated for the population of all training set data with the same geometry and wavelength. Both the training set and the observations go through this pre-processing standardization process. We found that the range of our standardized training set values for DoLP is roughly four time larger than that of $R_I$. This means that, relative to measurement uncertainty, DoLP is approximately that much more sensitive to the

parameters we vary in our training set than $R_I$.

## 3.3   Neural Network Architecture and Training

To handle the order-of-magnitude increase in the of the new input layer, we have been pushed to develop a deeper network architecture. The new network, shown in figure 3, consists of four subsequent hidden layers, each with $1,024$ nodes. This deep architecture contains more parameters that need to be trained, and as a consequence our approach to training has also changed.

In our previous work, we used a pure stochastic back-propagation method that updated the weights in the hidden layer after each training sample. This network is instead trained using a mini-batch method, where a batch of samples (128) is trained and the hidden layer weights are only updated after after each batch has been processed. In this architecture, following each hidden layer there is a batch normalization (BN) layer applied to the outputs of the layer. The purpose of the BN layer is to increase the stability of neural network, by subtracting the batch mean and dividing by the batch standard deviation. By applying this

transformation, it keeps the inputs into the subsequent activation layer stable (not too high and not too low), maintaining the





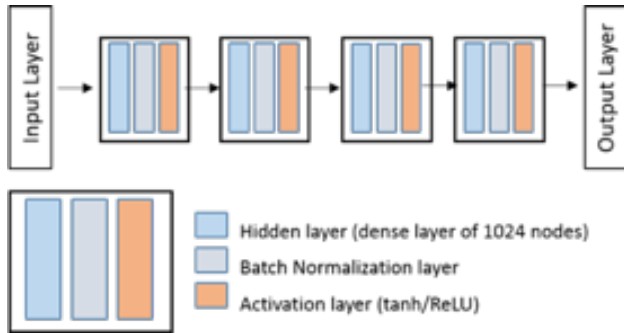

**Figure 3.** Architecture of the operational NN scheme, used for the retrieval of ORACLES 2016-2017 RSP measurements. The network contains four subsequent hidden layers, as detailed in the text.

mean activation close to $0$ and the activation standard deviation close to $1$, to help in the network training convergence. In the activation layer, we make use of either a hyperbolic tangent ($\tanh(\theta)$) or rectified linear unit function ($\mathrm{ReLU}(\theta)$), the latter of which is zero for all negative inputs and linearly increases for positive inputs. The $\tanh$ activation is widely used as the standard in NN literature (e.g. LeCun et al. (1989, 1998)), while $\mathrm{ReLU}$ is gaining more popularity recently due to its simplicity its ability

to greatly accelerate the convergence of stochastic gradient descent (SGD) algorithms and their variations (Krizhevsky et al., 2012). In Segal-Rozenhaimer et al. (2018), we did not notice a large difference between these two activation functions, but for this larger NN we find differences in retrieval performance that we will discuss in detail in section 4. Finally, the output layer activation function is linear, with a loss function defined as the mean square error (MSE) and results in the predicted values of $\tau$, $r_\mathrm{e}$ and $v_\mathrm{e}$.

During the training process, the input vector, which is pre-processed as detailed in the former section, is further scaled to have values between $-1$ and $1$, prior its ingestion into the network, to allow better convergence during training. Also, the training process is being regularized by adding Gaussian noise to the input layer during the training phase. The optimization algorithm used here is Adam (Adaptive moment estimation), implemented within the Keras python API (Chollet, 2017) with a TensorFlow backend (a system for large scale machine learning) (Abadi et al., 2016). In comparison with the classic stochastic

gradient descent (SGD) optimization algorithm, Adam is computationally efficient, has little memory requirements, and is well suited for problems that are large in terms of data and/or parameters (Kingma and Ba, 2014). The network is trained with a learning rate of 0.0001 using the "mini-batch" method to complete an epoch, while the number of epochs per training scenario was 100. Following training, the network was evaluated using an evaluation dataset consisting of a subset of training set data that was set aside during the training phase. Taking the network trained for ORACLES 2016 using tanh activation as an

example, comparison with the evaluation dataset resulted in correlations of $0.999$, $0.987$, and $0.941$; absolute biases of $0.016$, $0.044\,\mu\mathrm{m}$, and $0.094$; RMSE of $0.021$, $0.076\,\mu\mathrm{m}$, and $0.16$ each for $\tau$, $r_\mathrm{e}$, and $v_\mathrm{e}$ respectively. The results for $\tau$ and $r_\mathrm{e}$ are quite promising, but the RMSE in the $v_\mathrm{e}$ evaluation after training is enough to span the possible state space —an indication that this network cannot adequately retrieve $v_\mathrm{e}$.





**Table 3.** Correlations between existing RSP retrieval of $r_e$ and $\tau$ with raw NN retrieval output with different activation functions ($\mathrm{ReLU}(\theta)$ vs. $\tanh(\theta)$). Note that the p-values for all of these comparisons are much less than 0.05.

| | $r_e$ | | $\tau$ | |
|---|---|---|---|---|
| | Activation Functions | | | |
| Retrieval Correlations | $\mathrm{ReLU}(\theta)$ | $\tanh(\theta)$ | $\mathrm{ReLU}(\theta)$ | $\tanh(\theta)$ |
| Nakajima-King – (NJK) | 0.77 | 0.77 | 0.98 | 0.81 |
| Parametric Polarimetric – (PP) | 0.64 | 0.76 | 0.98 | 0.81 |
| Rainbow Fourier Transform – (RFT) | 0.60 | 0.67 | 0.98 | 0.81 |

## 4   Results

### 4.1   Initial Output and and Post-processing

The output from the network initially reveals some issues that still need to be addressed. Our approach to evaluating the behavior of the initial output layer results is to explore comparisons to the RSP retrievals. As indicated in section 2.3, we are

particularly interested in the RSP PP retrieval comparison as this provides the most consistent retrieval results in conditions with varying cloud inhomogeneity and in the presence of above cloud aerosols. Overall, this comparison revealed correlations for the $r_e$ retrievals are lower ($R \approx 0.7$) than for the $\tau$ ($R \approx 0.9$) retrieval irrespective of the network activation function used. An interesting finding of this initial analysis was that networks using different activation functions produced different behaviors for the retrievals of $\tau$ than they did for $r_e$. This is demonstrated in table 3 using the ORACLES 2016 network and data, where

the correlations for $r_e$ retrievals improve for networks using a $\tanh$ activation function, while in contrast $\tau$ retrievals have improved correlations for networks using a $\mathrm{ReLU}$ activation function. This behavior is a symptom of a feature we observed - rather than being linearly related to the PP retrieval of $\tau$, the $\tanh$-based $\tau$ retrieval demonstrated a non-linear or logarithmic dependence with increasing $\tau$. A similar behavior was exhibited for the PP retrieval of $r_e$ and the $\mathrm{ReLU}$-based $r_e$ retrieval. As a consequence, throughout the rest of this study we will perform retrievals for $r_e$ and $\tau$ with two different networks, for $r_e$

we make use of the $\tanh$ network and for $\tau$ we make use of the $\mathrm{ReLU}$ network. This approach will be further discussed in section 4.2.

Beyond simply evaluating correlations, the raw output of the network exhibits clear linear offsets when compared to the other RSP retrievals. In particular we emphasize this behavior for the PP retrievals in figure 4. This linear bias was absent during our training set validation exercise in section 3.3, implying that this systematic offset is consequence of differences between

training set and observational data. Despite this linear bias, the high correlations of these retrievals imply that the NN retrieval is otherwise generally performing correctly.The source of this bias is still an open question, and in the framework of NN it is difficult to diagnose the source of this kind of error. In particular we expect that this bias is an expression of a difference





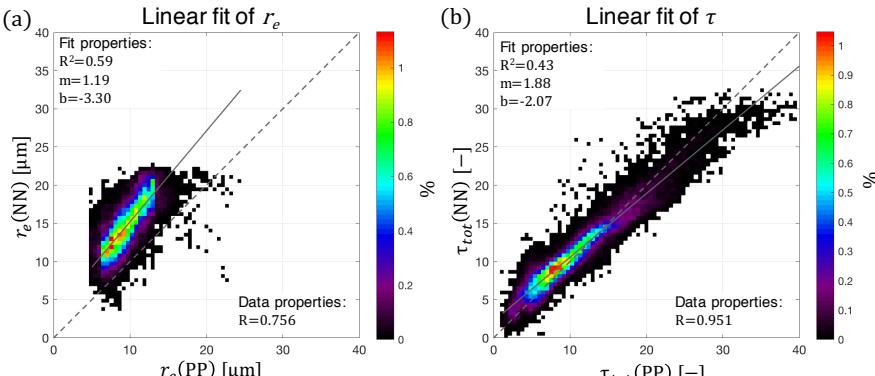

**Figure 4.** Density regression plots demonstrating behavior of raw NN retrieval output for $r_e$ (tanh network) and $\tau$ (ReLU network). The dashed gray line in each plot indicate the 1:1 line, while the solid gray line indicates the linear best fit line for the dataset.

between the assumptions built into the network training dataset that differ from the observation dataset. We will discuss the possible sources of these differences in section 5.

Given the high correlations, linearity of the initial output, and results from the network training evaluation, we believe the linear offsets of these regressions are artifacts. To correct the persistent linear offset of the NN retrievals we apply a linear
scaling to them and creating what we refer to as our adjusted NN retrieval product. Ideally and in principal this could be done with a small validation dataset that is not related to the retrieval products we hope to later compare our results to. Without an external dataset to scale to, we have decided to linearly scale the NN retrievals to a subset of 10% of the total population of RSP data (in this example for the ORACLES 2016 dataset). This avoids explicitly fitting all NN retrievals via fitting. This dataset is then regressed against the corresponding $\tau(\mathrm{PP})$ and $r_e(\mathrm{PP})$ retrievals to obtain two linear correction terms, the offset bias ($b$)
and the scaling bias ($m$),

$$x_{\mathrm{NN}} = m x_{\mathrm{PP}} + b, \tag{12}$$

$$\hat{x}_{\mathrm{NN}} = \frac{1}{m}\left(x_{\mathrm{PP}} - b\right). \tag{13}$$

Where $\hat{x}_{\mathrm{NN}}$ corresponds to a linearly adjusted neural network retrieval output, $m$ is the scale correction and $b$ is the linear offset correction. After the components of this linear adjustment are determined using the sub-set of PP retrievals the NN
retrievals are adjusted (i.e., the $\hat{x}_{\mathrm{NN}}$ product is created). Finally, this adjusted NN retrieval product can be compared to the other retrievals using the full RSP dataset – including the portion that was excluded from this correction definition exercise. The application of this linear correction does not influence the correlation of the retrievals; however, it does result in lower mean and RMSE biases for this ORACLES 2016 example shown in figure 4. For $r_e$ the mean bias is reduced to $0.023\,\mu$m and the RMSE is $1.74\,\mu$m, whereas for $\tau$ the mean bias is $-0.050$ and the RMSE is $1.82$. Further discussion of the behavior of





the adjusted NN datasets are separated into section 4.2 and section 4.3, which each discuss and highlight behaviors of the NN retrieval for the ORACLES 2016 and 2017 datasets respectively.

## 4.2 Results for ORACLES 2016

From a number of perspectives, the ORACLES 2016 campaign data is easy to work with, RSP was flying on a dedicated remote sensing platform (NASA high-altitude ER2), there were prevalent observations of clouds, and data availability was often not an issue. As a consequence, the dataset analyzed here is large —including six days of flights with $N = 72,542$ retrievals that pass all of the analysis filter criteria introduced in section 2.2.

The overall statistics of retrievals during ORACLES 2016 highlight features and challenges for the development of the NN retrieval. At first glance, the retrieval probability distribution functions (PDF, [%/bin units]) in figure 5 reveal that all of the RSP cloud retrievals are similar to one another – with droplet sizes that are small ($\hat{r_e} \approx 10$), and optical thicknesses are largely moderate ($\hat{\tau} \approx 7$) and relatively few occurrences of thicker clouds ($\tau > 30$). The standard RSP cloud retrievals exhibit some similar differences and behaviors to those highlighted in section 2.3, specifically that the NJK retrieval is shifted toward larger droplet sizes than the other two methods. An evident feature of the NN retrieval PDF is that there is some clustering occurring near discrete values associated with the training set grid (shown below each PDF as defined in table 1), as demonstrated by the NN training grid bins shown below each PDF. This effect is particularly evident in the $\tau$ histogram where peaks in the PDF appear near training set grid points (allowing for some shifting associated with the post-processing correction). The overall shape of the NN retrieval distributions resemble that of the other RSP retrievals, although the NN retrievals of $r_e$ appear to be slightly more broadly distributed.

A closer examination of the comparison of NN retrievals and the PP retrievals is required to reveal if the NN retrieval exhibits any systematic dependence on different retrieval populations. This is accomplished using the joint-density regression of the NN retrievals against each of the PP retrievals shown in figure 6. Comparisons of the NN retrievals to the RSP PP retrievals reveal mean biases for $r_e$ and $\tau$ of $0.023\,\mu\mathrm{m}$ and $-0.050$ respectively, with RMSE for $r_e$ and $\tau$ of $1.74\,\mu\mathrm{m}$ and $1.82$ respectively. In the case of the $r_e$ retrieval, the NN retrieval appears to miss the large $r_e(\mathrm{PP})$ retrieval population above $15\,\mu\mathrm{m}$ and there is much more variability in small $r_e(\mathrm{NN})$ retrievals in part because there are no $r_e(\mathrm{PP})$ retrievals below $5\,\mu\mathrm{m}$. Whether or not such small $r_e(\mathrm{NN})$ results are reasonable remains an open question as this regime is often excluded from look-up table datasets, whether for sensitivity reasons (NJK has multiple solution issues) or simply because they are not expected to be common.

A flight track time-series is useful for emphasizing how the observed spatial variability of the NN retrieval behaves relative to the other retrieval products. The example flight track time-series in figure 7 reveals that there is clearly good match-up between cloud optical thickness and effective radius retrievals. The statistics of this time series show improvement relative to our previous study. In particular the new NN exhibits a retrieval of $r_e$ that is significantly improved relative to Segal-Rozenhaimer et al. (2018) – with correlations between $r_e(\mathrm{NN})$ and $r_e(\mathrm{PP})$ of $0.587$ and an RMSE between $r_e(\mathrm{NN})$ and $r_e(\mathrm{PP})$ of $1.74\,\mu\mathrm{m}$ . The new network performs just as well on the $\tau$ retrieval as our previous study – with correlations $\tau(\mathrm{NN})$ and $\tau(\mathrm{PP})$ at $0.951$ and an RMSE between $\tau(\mathrm{NN})$ and $\tau(\mathrm{PP})$ of $1.842$.

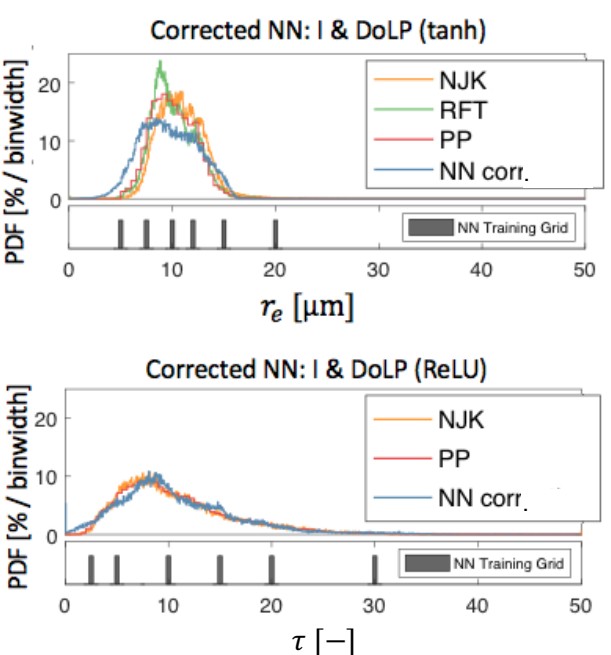

**Figure 5.** PDF's of RSP retrievals from flights during ORACLES 2016.

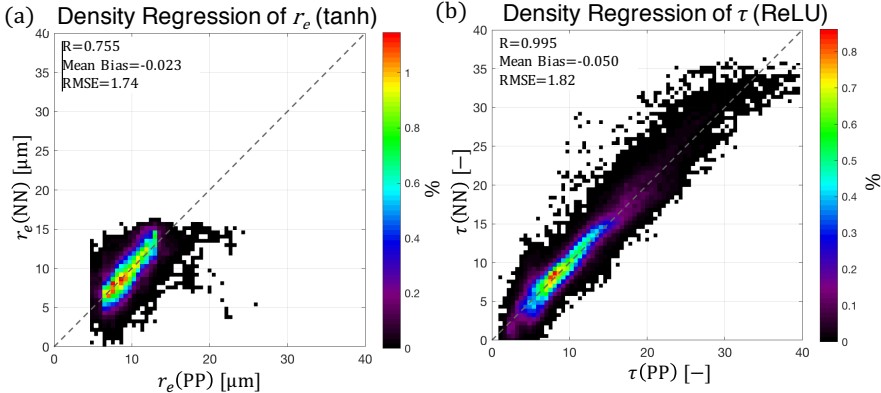

**Figure 6.** Density regression plots comparing all of the ORACLES 2016 NN retrievals (y-axis) and PP retrievals (x-axis) of $r_e$ (panel a) and $\tau$ (panel b). The density of the joint-histogram is shown in a linear scale indicating the percentage of the retrieval population within each bin. A dashed-line is shown also plotted that indicating the $1:1$ line.

Additionally, we found that the NN retrieval demonstrated some dependence on untrained variables that influence the observational dataset. First, we found a dependence to the fixed cloud top height assumption ($H = 1\,\mathrm{km}$) that was made for the ORACLES 2016 training set. This was revealed by comparing our percent retrieval bias (relative to $r_e(\mathrm{PP})$) to the HSRL-2





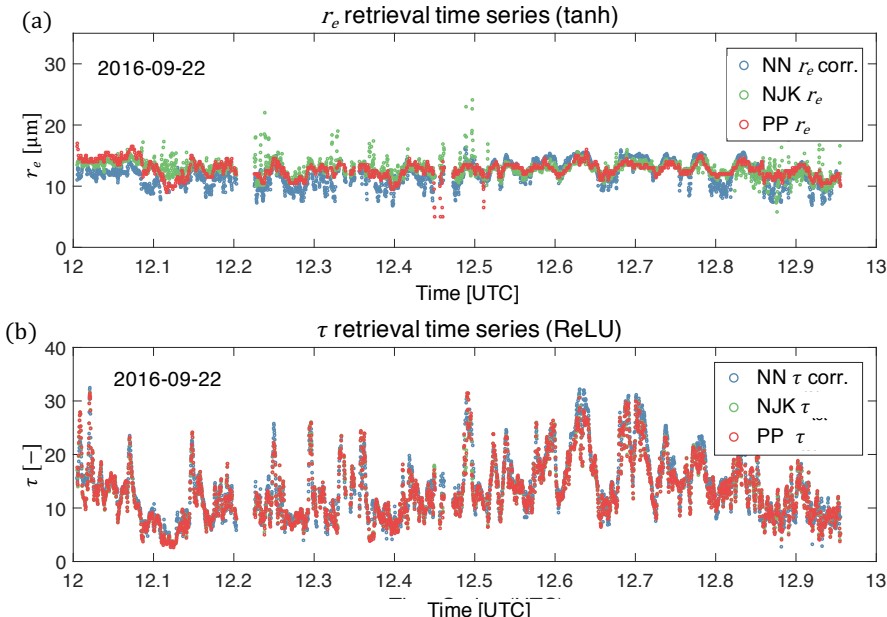

**Figure 7.** A selected time series of $r_e$ (panel a) and $\tau$ (panel b) retrievals from NN (blue), NJK (green), and PP (red) methods.

cloud top height product, as shown in figure 8. There is clear covariability between the HSRL-2 cloud top height and the bias between $r_e(\mathrm{NN})$ and $r_e(\mathrm{PP})$. In this example, it appears as though the cloud top height variation could be associated with a $\pm 20\%$ variation in the percent retrieval bias depending on the relative error in the cloud top height assumption. A second sensitivity we observed influenced the $\tau(\mathrm{NN})$ retrieval in the presence of above cloud aerosols. While there was no clear functional

dependence, we did observe a handful of cases where the HSRL-2 above cloud aerosol optical thickness ($\tau_{\mathrm{ACA}}$) was weakly correlated with a reduction in the $\tau(\mathrm{NN})$ retrieval.

### 4.3 Results for ORACLES 2017

The ORACLES 2017 campaign data presented a more difficult dataset to work with. Observations that lacked SWIR data and fewer collocated HSRL-2 observations on the days when SWIR data was available reduced the amount of useful intercompar-

10 ison data. As a consequence, the dataset analyzed here is smaller than 2016 —including four days of flights with $N = 18,159$ retrievals that pass all of the analysis criteria discussed in section 2.2. This difficulty also presented an opportunity to test the behavior of a NN retrieval trained without SWIR data at all. To accomplish this an alternate version of the 2017 ORACLES NN was developed that was trained without SWIR data. Then, using the same observational dataset from ORACLES 2017 (namely the data with SWIR observations) was input into both networks either with or without SWIR data accordingly. As

shown in table 4, the behavior of the NN that excluded SWIR data behaved quite differently than the network trained for SWIR observations.



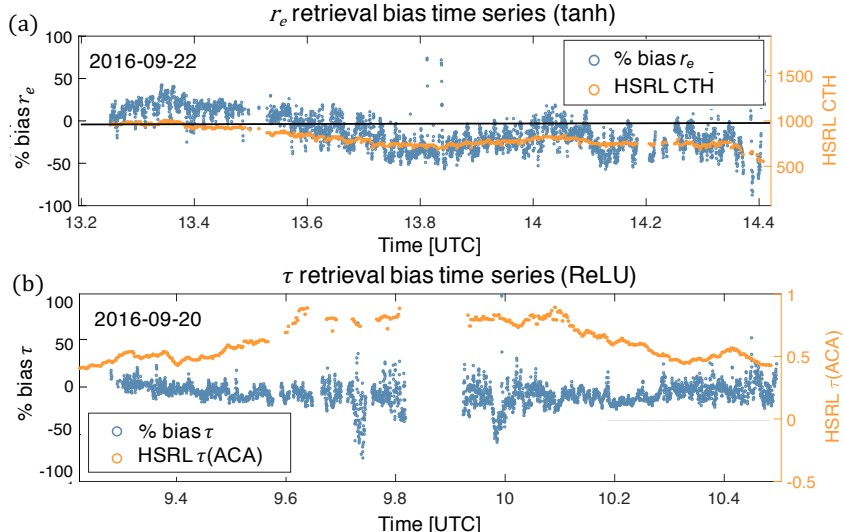

**Figure 8.** Panel (a) is a time series of the percent bias of $r_e(\text{NN})$ with respect to $r_e(\text{PP})$ plotted along with the accompanying HSRL-2 CTH time series overtop. The solid line going through the center of the figure indicates both zero bias and the CTH assumption of $1\,\text{km}$. Panel (b) shows the time series of the percent bias of $\tau(\text{NN})$ with respect to $\tau(\text{PP})$ plotted along with the accompanying HSRL-2 $\tau_{\text{ACA}}$. Note that these biases are shown for datasets on different days.

**Table 4.** Comparison of initial NN output (uncorrected) to PP retrievals during ORACLES 2017. All $r_e$ retrievals are for a tanh-based network, and all $\tau$ retrievals are for a ReLU-based network.

|  | $r_e$ w/ SWIR | $r_e$ w/o SWIR | $\tau$ w/ SWIR | $\tau$ w/o SWIR |
|---|---|---|---|---|
| $R$ | $0.54$ | $-0.325$ | $0.782$ | $0.903$ |
| RMSE | $4.77\,\mu\text{m}$ | $5.86\,\mu\text{m}$ | $5.78$ | $3.10$ |

As might be expected, the exclusion of SWIR reflectances has a significant detrimental impact on the NN retrieval of $r_e$. Compared to the moderate correlation and RMSE error of NN retrievals with SWIR data ($R = 0.54$ and $\text{RMSE} = 4.77\,\mu\text{m}$) NN retrievals of $r_e$ without SWIR data have a very poor correlation and RMSE ($R = -0.325$ and $\text{RMSE} = 5.86\,\mu\text{m}$). This behavior is likely attributable to the loss of information content in the SWIR bands, which are strongly absorbed by liquid water droplets and as a consequence are more sensitive to droplet cross section (and therefore $r_e$) than the other spectral bands.

Perhaps unexpectedly, the exclusion of SWIR reflectances in the training and observation dataset improves the correlation and RMSE between $\tau$ NN retrievals and the other standard RSP retrievals of $\tau$. However, as the comparison of the corrected datasets in figure 9 reveals, this story is slightly more complex and nuanced. On the one hand, the histogram regressions clearly show that the $\tau$ retrieval *with* SWIR reflectances (figure 9(a)) is more broadly distributed and exhibits a non-linear dependency





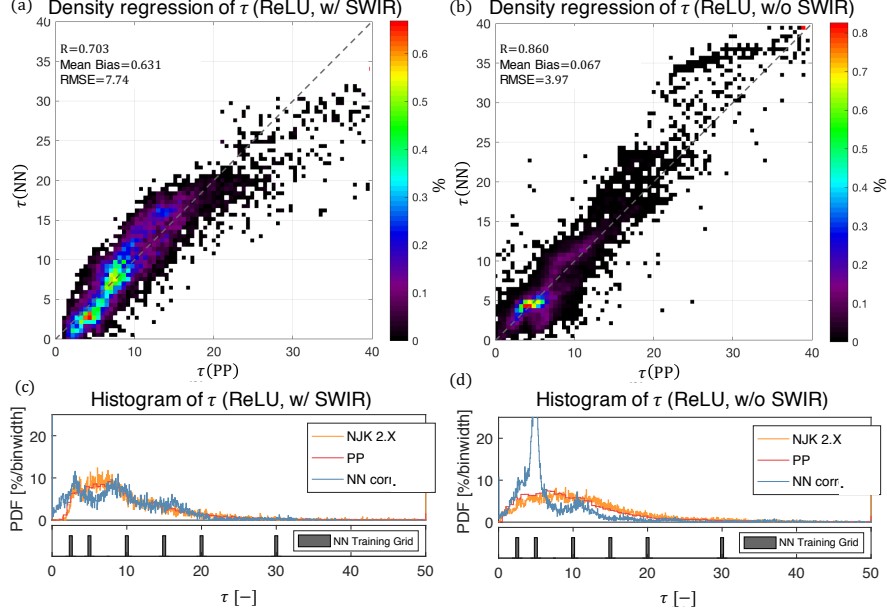

**Figure 9.** Comparisons of corrected NN retrievals against the PP retrievals of $\tau$ for the ORACLES 2017 dataset. The left two panels focus on the NN retrieval with SWIR reflectances while the right two panels focus on the NN retrieval without SWIR reflectances. The top panels are histogram regressions while the bottom panels show the corresponding 1-D histogram of the NN retrieval (below which the training grid is shown).

that gets exacerbated for large $\tau$, while the $\tau$ retrieval *without* SWIR reflectances (figure 9(b)) is more tightly distributed and more linearly correlated. This seems to confirm the relationship indicated in the bulk statistics of table 4. On the other hand, the 1-D histograms reveal that the distribution of the retrieval *without* SWIR reflectances (figure 9(d)) is mostly densely clustered around bin locations of the training set grid (indicated below as a bar plots). Thus, it appears as though a NN retrieval without

SWIR data may be possible – if the training set had a higher density of grid points.

Another interesting finding regarding the NN retrievals in ORACLES 2017 stems from how it compares to the other standard RSP retrievals. In addition to comparison with the PP retrieval as we have highlighted up until now, we have also evaluated comparisons of the NN retrievals against the NJK retrieval. At first glance, the coarse statistical comparison of the initial NN output to the NJK retrieval in table 5 reveal similar results to those compared to the PP retrieval in table 4. However, at closer

inspection the $r_e$ retrieval comparison to NJK (with SWIR) has correlations and RMSE that are moderately better than the results from the PP comparison.

Apparently, the ORACLES 2017 NN retrievals of $r_e$ do not compare as well to the PP retrieval as the results in ORACLES 2016. The histogram regressions in figure 10 reveal this clearly where the comparison to the PP retrieval (panel a) shows a clear non-linearity whereas the comparison to NJK retrieval (panel b) is more linear. Each comparison has similar RMSE but

there are also important differences in the distribution of retrievals. In particular, the nonlinear behavior of the comparison to





**Table 5.** Comparison of initial NN output to NJK retrievals during ORACLES 2017. All $r_e$ retrievals are for a tanh-based network, and all $\tau$ retrievals are for a ReLU-based network. .

|  | $r_e$ w/ SWIR | $r_e$ w/o SWIR | $\tau$ w/ SWIR | $\tau$ w/o SWIR |
|---|---|---|---|---|
| $R$ | 0.627 | 0.536 | 0.785 | 0.908 |
| RMSE | $3.78\,\mu m$ | $4.51\,\mu m$ | 5.61 | 3.08 |

the PP retrieval is reminiscent of the biases shown previously during the comparison of NJK and PP to one another directly in figure 2, where there were also large high-biases in the NJK retrieval for small and large droplet size regimes. Previously, we concluded that this difference was associated with thin ($\tau < 3$) or broken clouds. The increased relative occurrence of thin and broken clouds that characterized the observations made during ORACLES 2017 appears to be the primary source of this behavior. This population of clouds is most susceptible to biases that are coupled to spatial resolution – specifically unresolved cloud inhomogeneity and resolved 3-D radiative effects. These effects are known to have a more severe influence on the NJK $r_e$ retrieval than on the PP retrieval of $r_e$ (Miller et al., 2016, 2018). Interestingly, because the NN retrieval is ingesting reflectances that may be biased by these effects the NN retrieval more closely resembles the results of the NJK retrieval rather than the PP retrieval. This appears to indicate, at least for the ORACLES 2017 dataset, that the NN places is influenced strongly by biased total reflectances, particularly for the optically thin clouds that were often observed.

Looking at the flight track time-series the observed spatial variability of the ORACLES 2017 NN retrieval in figure 11 reveals some similarities to the ORACLES 2016 cases examined previously. In particular, the spatial variability of the ORACLES 2017 NN retrievals of $\tau$ appears similar to the results shown previously in figure 7. However, looking more closely at the $r_e$ time series reveals there is a clear deviation of both the NN and NJK retrievals around a gap in the cloud at approximately 10.5 UTC – a behavior not observed in the PP retrieval. This is evidence that cloud inhomogeneity and thin clouds ($\tau < 3$) are indeed the source of the biases observed in both the NN and NJK retrievals in this dataset. Additionally, there are notable deviations of the NN retrieval of $r_e$ away from other RSP retrievals in the proximity of steep increases or decreases in $\tau$ (e.g., around 10.7 or 10.85 UTC). This behavior could be a consequence of stronger 3-D radiative effects in shortwave spectral bands that are not used by the NJK retrieval but are a part of the NN framework.

While the ORACLES 2016 NN retrievals exhibited correlation to some untrained variables like cloud top height and above cloud aerosol optical thickness the results from ORACLES 2017 did not reveal any meaningful correlations with ACA optical thickness. Additionally, there was no clear sensitivity of the new network to cloud top height. Indicating that training with the flight altitude as a variable during training had a positive impact on the retrieval outcomes for this dataset.

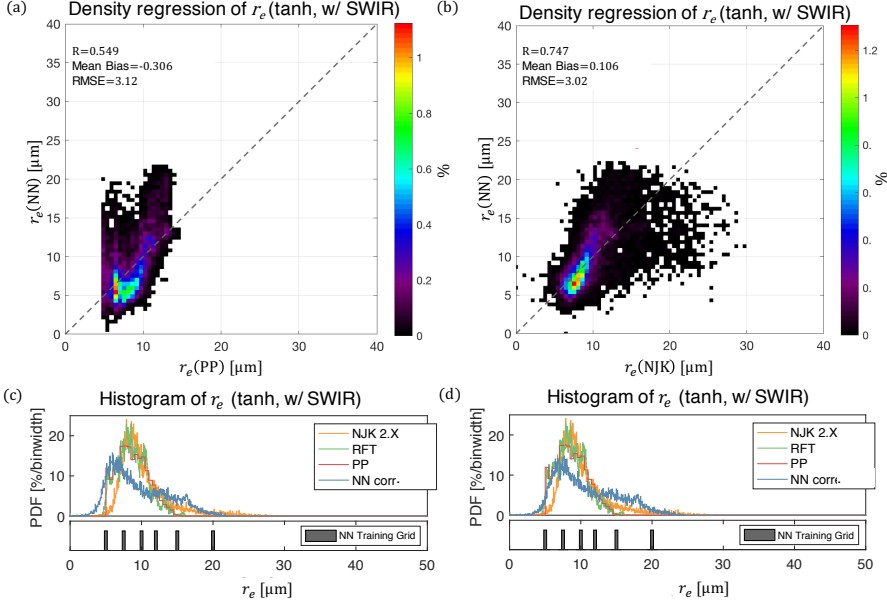

**Figure 10.** Histogram regressions of $r_e$ retrievals that compare NN to PP retrievals (panel a) and NN to NJK retrievals (panel b). Below are the 1-D histograms of the different NN corrected retrievals using different reference retrievals (PP, panel c; NJK panel d) for the linear correction of the initial NN output.

## 5   Summary and Discussion

Overall, the results of this study demonstrate that a multiangular polarimetric neural network cloud property retrieval can produce results that are statistically similar to other existing RSP cloud retrievals. Defining the input layer of the NN required careful consideration of the particularities of multiangular polarimetric data. For example, we found that appropriately

weighting of observations that have vastly different uncertainties was quite important because total ($I$) and polarized (DoLP) observations differ from one another by an order of magnitude in both value and uncertainty. Additionally, we constructed a deeper network architecture and created a more efficient network that could operate on the entire observation vector itself, rather than on a reduced input vector. After making these input vector decisions, each retrieval was performed using a separate network architecture, each giving the best results for the given variable ($r_e$ or $\tau$). Specifically, we found that networks using

different activation functions performed better for retrievals of $\tau$ and $r_e$ – namely, the network using $\tanh$ for $r_e$ retrievals and the network using $\mathrm{ReLU}$ for $\tau$ retrievals. In addition to using different networks for $r_e$ and $\tau$ retrievals, the inherent differences between the ORACLES 2016 and 2017 datasets required us to develop different networks for each year that were built using training data tailored for the observation conditions. This effort was complicated by the fact that the two datasets differed significantly, with many more broken and inhomogeneous clouds present in the ORACLES 2017 dataset. This presented a

challenge for the NN and other RSP retrievals, but also an opportunity for us to learn how the NN behaved in a larger variety of conditions.



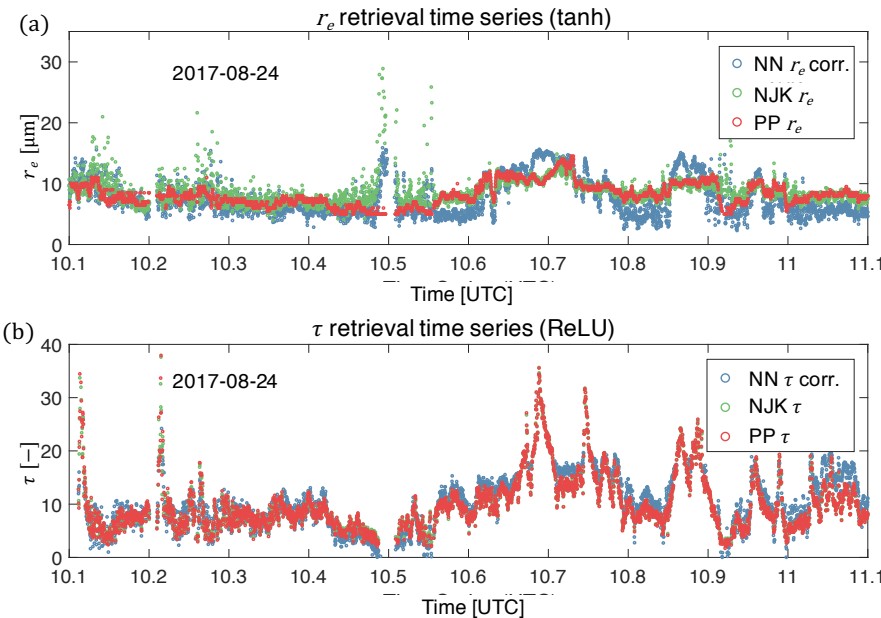

**Figure 11.** A selected time series of $r_e$ (panel a) and $\tau$ (panel b) retrievals from NN (blue), NJK (green), and PP (red) methods.

As discussed in 4.1, the initial output of the NN exhibited a clear systematic linear offset relative to the other RSP standard retrievals. This was especially true for the $r_e$ retrieval, which had an offset bias of about $3\,\mu$m. To create a dataset that was more consistent with the other RSP retrievals we arbitrarily linearly corrected the NN retrieval datasets by linear regression to the RSP PP retrieval for a limited sampling of retrievals. Afterwards, the linear correction was applied to the full dataset the results

were again compared to other RSP cloud retrievals using the correlations and RMSE statistics as a meaningful evaluation of the retrievals quality. However, the source of the linear offset bias likely stems from a difference in the data used in the training set and the observations made by the instrument. The simplest explanation for this difference is associated with above-cloud gaseous absorption that was not modeled in the NN training sets. The absorption of well-mixed (e.g., $CO_2 and CH_4$) and trace gases (e.g., water vapor (WV), $NO_2$ and $O_3$) can vary significantly within some of the spectral bands of RSP – of particular

note is strong absorption by $CO_2$ and $CH_4$ in two SWIR bands where much of the sensitivity to cloud droplet size information is contained. Additionally, the absorption of these gases also increases with increasing view angle as the light scattered to the detector passes through a longer atmospheric path at oblique viewing angles. To test the impact atmospheric absorption we re-examined a pair of cases of atmospherically-corrected RSP data and compared them to our original NN retrievals time series data from 2016 and 2017 shown in figure 7 and figure 11. The modeled atmosphere was built using RSP retrievals of above

cloud WV, in addition to MERRA-2 reanalysis column $NO_2$, $O_3$ and subsequent assumptions regarding well-mixed gases and vertical profiles based on the US standard atmosphere (National Aeronautics and Space Administration et al., 1976). Cloud top height measurements from HSRL2 were used to define cloud top and subsequently calculate the above cloud impact of to the





**Table 6.** A summary of the properties of the archived RSP NN retrieval. All $r_e$ retrievals are for a tanh-based network, and all $\tau$ retrievals are for a ReLU-based network. The linear corrections applied to the initial NN output are recorded here so that one could replicate results presented in this paper.

| Retrieval | Activation Function | ORACLES 2016 | | ORACLES 2017 | |
|---|---|---|---|---|---|
| | | Linear Offset (b) | Scaling Factor (m) | Linear Offset (b) | Scaling Factor (m) |
| $r_e [\mu m]$ | tanh | 3.2854 | 1.1903 | 4.1348 | 0.93073 |
| $\tau [-]$ | ReLU | -2.1733 | 1.8922 | -0.08834 | 1.6781 |

absorption of the two-way transmitted reflectance. The $r_e$ retrievals from the initial NN output (not linearly adjusted) that are obtained using the atmospherically corrected reflectances for ORACLES 2016 and ORACLES 2017 networks are compared to the original scaled NN retrievals and the polarimetric RSP retrieval in figure 12. It is evident that the atmospheric correction largely serves to reduce the $r_e$ retrieval globally to a value that is more in line with the polarimetric retrieval —cutting the offset bias nearly in half. This is likely due to the correction for absorption in the SWIR bands as well as the correction to the angular distribution of reflectance due to large view-angles having significantly more absorption. It is also good to note the atmospheric correction has very little impact on the NN retrieval of $\tau$, which did not have a significant offset bias. It should also be noted that after correcting the observational input, the tanh-based network retrieval of $\tau$ improved markedly – indicating that this activation function may be more useful than we found previously. These initial results are promising because they largely do not change the variability in the time-series and as a consequence validates our linear correction approach. As a consequence, atmospherically corrected reflectances was used for the dataset available in our archive (refer to the data availability statement).

# 6 Conclusions

Comparisons of the NN retrieval to the existing RSP cloud retrievals during ORACLES revealed reasonable results. In particular, the ORACLES 2016 dataset showed comparisons of neural network retrievals (NN) to the RSP polarimetric retrievals (PP) that had correlations for $r_e$ and $\tau$ of $R = 0.756$ and $R = 0.950$ respectively, while the RMSE for $r_e$ and $\tau$ were $1.74 \mu m$ and $1.82$ respectively. The results of this comparison are of similar quality to the comparison of the standard RSP PP and NJK retrievals to one another in figure 2. In contrast to these results, the ORACLES 2017 dataset fared poorly, with correlations and RMS errors of NN retrievals of $r_e$ ($R = 0.54$ and $\mathrm{RMSE} = 4.77 \mu m$) and $\tau$ ($R = 0.785$ and $\mathrm{RMSE} = 5.61$) that were much worse. Though, based on the comparisons of the standard RSP PP and NJK retrievals of $r_e$ this was to be expected due to the increased prevalence of optically thin ($\tau < 3$) clouds observed in the ORACLES 2017 data. As a consequence, the NN retrievals of $r_e$ during this year more closely resemble the systematically high-biased NJK retrievals. It is however, surprising that the $\tau$ retrieval performed so poorly for this dataset. It appears to be the result of a strong non-linear behavior with increasing $\tau$. We found that if we attempted to retrieve $\tau$ using an input vector that excluded the SWIR data, then the $\tau$ retrieval statistics improved significantly ($R = 0.908$ and $\mathrm{RMSE} = 3.08$). However this SWIR-free NN retrieval exhibited

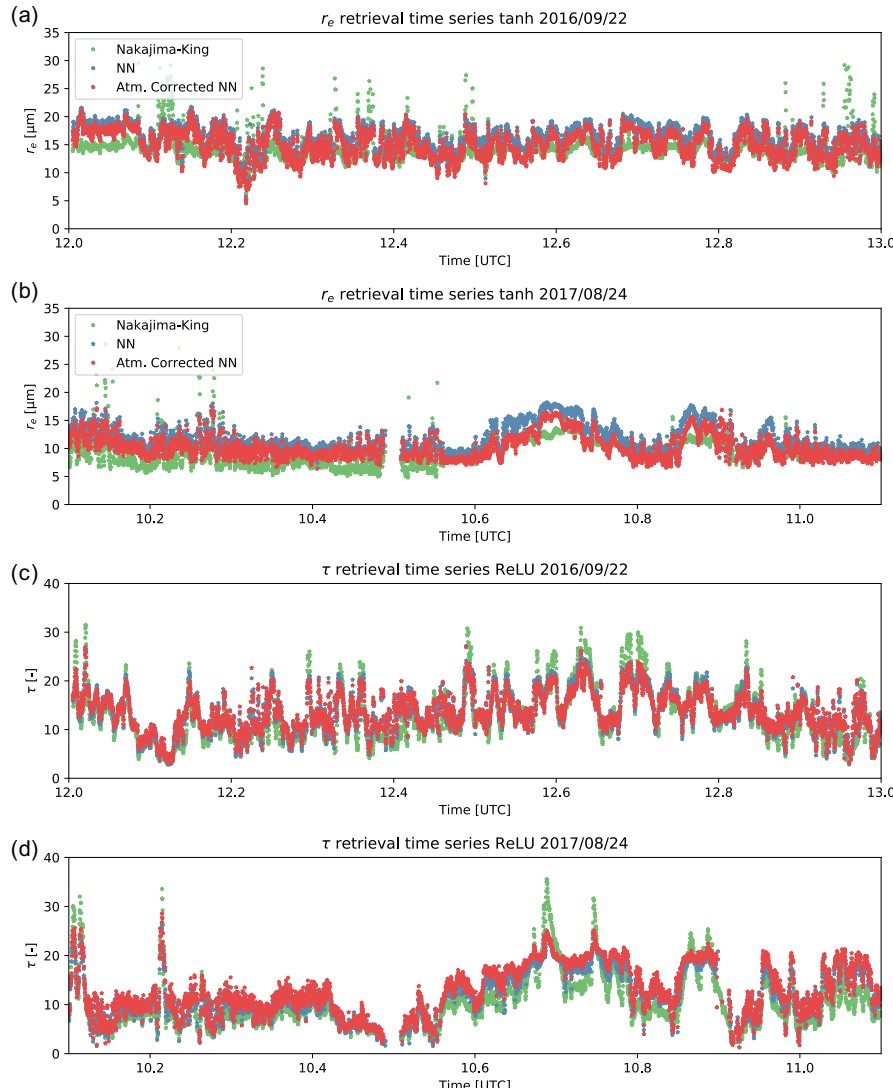

**Figure 12.** Time series comparisons of NJK retrievals (green), uncorrected NN retrievals (blue), and NN retrievals using atmospherically corrected reflectances. The top two panels focus on $r_e$ retrievals from ORACLES 2016 (top) and 2017 (below) while the lower two panels present the $\tau$ time series from ORACLES 2016 (above) and 2017 (bottom).

an undesirable training set bin-seeking behavior that may possibly be avoided if we trained with a denser grid of training set variables.

The NN was trained using a synthetic dataset that made some significant assumptions about the types of scenes that would be observed. The first type of assumption relates to the structure of the forward model itself (i.e., assuming that clouds are
5    plane-parallel and internally homogeneous to simplify to a 1-D radiative transfer problem). As a consequence, the cloud





is assumed to be vertically homogeneous, which could cause issues due to the different vertical information contained in polarized reflectances which scatter from a shallow layer at the cloud top while the total radiances contain information from deeper within the cloud. While the second type of assumption is about the state of the atmosphere itself (i.e., using a fixed cloud top height). Another example of this type of assumption is that the observations in the training set exclusively consider

the presence of clear cloudy scenes, with no aerosols above the cloud. We cannot do much about the first type of assumption, as cloud retrievals are subject to the possible influence of inhomogeneity and 3-D radiative effects –which usually has a greater impact on total reflectance-based retrievals than on polarimetric retrievals (Miller, 2017). On the other hand, the second type of assumption is something that can be further explored in future studies by incorporating a more complete description of the atmospheric state in the training dataset. We experimented with one such assumption of this type by training the network

for ORACLES 2017 to account for variability in the separation between the Aircraft altitude and the cloud top height. As a consequence the retrievals for the ORACLES 2017 dataset did not demonstrate the same systematic bias in $r_\mathrm{e}$ as a function of cloud top height that we observed in the ORACLES 2016 dataset. We have not yet extensively tested how above cloud aerosols influence the results of the NN retrievals shown here. There was some indication in 8 that ACA could lead to slight low biases in $\tau$ retrievals compared to the RSP PP retrieval, though in that instance both retrievals of $\tau$ should be impacted by the ACA

layer.

The NN approach outlined here comes with some strengths and weaknesses that must be considered. One of the weaknesses of a NN approach is that it lacks a clearly traceable relationship between observations and retrievals. However, despite this, the NN retrieval was shown here to provide reasonable results, lending support to the idea that a NN retrieval could provide a quick reasonable first guess to more complicated and physically traceable retrievals (Di Noia et al., 2015). Another interesting

feature of the approach we have taken for the NN retrieval is that it makes full use of the large information content of multi-angular polarimetry, using all the total and polarized reflectances numerous wavelengths and viewing geometries in the same retrieval. This is unlike the other RSP retrievals, which typically make use of a limited wavelengths and either polarized or total reflectance observations. As a consequence of using both total and polarized reflectances, we found that the NN retrieval was sometimes behaving more like the PP retrieval (when the clouds were optically thick and more homogeneous) and sometimes

behaving more like the NJK retrieval (when clouds were thinner and less homogeneous). It is possible that this is an indication that the NN is detrimentally influenced by biases in total reflectances associated to 3-D radiative effects and unresolved cloud inhomogeneity that also impact the NJK retrieval. These biases in total reflectance can be most severe for thin and broken clouds. This brings us to another weakness, the analysis of this study hinges on the comparison of retrievals to other retrievals. Without a true reference, we only have comparisons between different approaches, which comes with its own caveats and

sources of bias. This is specifically important in the case when both retrievals are not considering a large component of the observed system (e.g., the presence of an above cloud aerosol layer).

In the future we intend to extend this NN retrieval study in a few aspects. First, we endeavor to redefine our approach to developing the training set. Rather than use a fixed grid training set as we did in this study, we would like to use an approach that is more flexible and allows us to train more in the regions of state space that occur most often. One of the ways to do

this is to implement importance or occurrence sampling in the training set. Importance sampling requires the user to define

the full distribution of a priori cloud and aerosol parameters in the state space. Then the user specifies a number of training samples desired and then a value is randomly sampled from the distributions of each of the atmospheric state variables leading to numerous unique combinations of atmospheric simulations. This results in a training set that more accurately represents the underlying state space of the observational dataset and avoids the binned retrieval issues we saw in some of the NN retrievals in

this study. Second, because we saw improvement in the NN results when we corrected for atmospheric absorption we intend to improve our approach to atmospheric correction – making the input observations more closely resembles the synthetic training dataset. Third, with our understanding of the cloud retrieval problem on firmer ground we hope to extend the NN retrieval approaches discussed here to the retrieval of above cloud aerosol properties. Finally, we would also like to demonstrate that this NN first guess can indeed accelerate a rigorous optimal estimation retrieval of above cloud aerosol properties by providing

an accurate a priori estimate of the retrieval space that should be explored. The lessons learned from this study can hopefully help in other applications of machine learning to remote sensing data. The full RSP NN retrieval product can be accessed from the DOI in the data availability statement below.

*Data availability.* The RSP NN cloud retrievals from ORACLES 2016 and 2017 field campaigns can be found at: (DOI will be created before final publication). All other RSP retrieval products can be found at: https://data.giss.nasa.gov/pub/rsp/.

*Author contributions.* DM, KK, and MS-R drafted the manuscript together. The network architecture defined and developed by MS-R, while the simulated RSP training dataset was generated by KK, and the input standardization and uncertainty scheme was created by DM. Primary analysis and data processing was handled by DM. Integration of RSP measurements, retrievals, and atmospheric correction in this study was supported by BC, MA, BvD, and AW. All authors contributed to the editing of the manuscript.

*Competing interests.* The authors declare that they have no competing interests.

*Disclaimer.* TEXT

*Acknowledgements.* The authors would like to acknowledge NASA funding and support through NNH13ZDA001N-EVS2 for the ORA-CLES project. We would like the thank the ORACLES ER-2 and P3 aircrew for its support in the field. Additionally, DM's research was supported by an appointment to the NASA Postdoctoral Program at the NASA Goddard Space Flight Center, administered by Universities Space Research Association (USRA) under contract with NASA. The authors are grateful to Sharon Burton, Johnathan Hair, and Richard

Ferrare of the NASA Langely HSRL-2 team for making their data available and providing feedback on our data.





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
