# Peer review of "Low-level liquid cloud properties during ORACLES retrieved using airborne polarimetric measurements and a neural network algorithm"

_Atmospheric Measurement Techniques, 2019_

## Referee Comment (RC1) · Anonymous Referee #1 · 14 Oct 2019

The Authors present a neural network algorithm to retrieve the microphysical properties of liquid water clouds from multi-spectral multi-angle polarimetric measurements such as those carried out by the RSP instrument. They trained a number of NN algorithms from synthetic data and applied them to RSP measurements acquired over the south-eastern Atlantic Ocean during the ORACLES campaigns held in 2016 and 2017. On these campaign data, they compared their NN retrievals to two other retrieval schemes, one using polarization and one based on a typical Nakajima-King radiance look-up table.

The results look partly encouraging and partly in need for further investigation. As I

will detail below, a number of design choices raise some questions (keeping the cloud height fixed in the generation of the training dataset, going for a neural network with an enormous number of hidden neurons, training on a relatively coarse grid for some parameters, an input scaling mechanism that is supposed to force the NN to give less weight to more uncertain inputs but looks somehow dubious to me). Therefore, I would invite the Authors to elaborate more on the rationale behind such choices. Furthermore, I am a bit surprised by the fact that the NN does not seem capable of retrieving effective variance, as work published earlier this year suggested that this is possible with multi-angle polarimetry, even with instruments that observe at fewer angles and wavelengths than RSP (i.e. POLDER).

Below are my detailed comments.

- P1, L1-2, I would suggest to replace "relate ... microphysical properties" with "retrieve cloud microphysical properties from multi-angular and multi-spectral polarimetric remote sensing observations"

- P3, L33-34, sentence "in a manner unbiased by prior understanding of that relationship". This is not entirely true, or at least needs rephrasing. If you use a forward model to generate the training dataset for your NN, the dataset will exactly capture "your prior understanding of that relationship". If, instead, you mean that by using a training dataset you don't impose a specific analytical parameterization on that relationship, then your statement is correct, but it needs to be expressed better.

- P7, L29. If you exclude SWIR channels, what is the use of radiance in estimating effective radius? Wouldn't it be better to train a specific NN only using polarized radiance, as done in Di Noia et al. (2019)?

- P8, L9. I guess "This screening criteria was" should read "These screening criteria were" (by the way, here I am assuming that you mean the two criteria involving HSRL-2 among those you listed above)

**AMTD**
- P12, L10-11. The reason for the choice of fixing the cloud top height at 1 km in the training dataset looks a bit puzzling. Why fix it and not change it just like the other parameters? After all, I guess cloud height is definitely going to have an impact at least on polarization at shorter wavelengths.

- P13, Table 1. What did you do with the parameters affecting the reflectance of the ocean surface, such as wind speed or ocean color? I guess this may well be important over optically thin or broken clouds.

- P13, L7, "principle" should read "principal"

- P14, L7-9. Standardization is just one of the typical pre-processing steps. Another one, just as typical, is to linearly scale the NN inputs to a specified range, such as [-1,1]. Together with a good initialization of the NN weights (which are typically initialized within the same interval), this choice makes the cost function for training easier to optimize.

- P14, L17-20. Usually the goal of standardization (or linear scaling) as a preprocessing step for NN training is to bring all the input and output variables to the same variability range, as this is usually beneficial to any gradient-based optimization problem (Nocedal and Wright, 1999, Chapters 2 and 4). If your method doesn't do that, then I would say it's questionable, as it shapes the cost function in a more unfavourable way for convergence. I am not even sure that your standardization method will help the NN to give less weight to reflectance, as the training process may adjust the neural network weights in a way that compensates for that. You should verify if this is the case by looking at the final values for the weights, and possibly at the derivatives of the NN output with respect to the input variables. In general, if one wants to force the NN to give less weight to certain inputs, I guess that one would have to act on the derivatives of the training cost function with respect to the weights (via regularization), rather than on how the input variability range is scaled.

- P14, Section 3.3. It would be interesting to see how PP, NJK and RFT perform on the

AMTD
synthetic dataset you used to evaluate the NN retrieval.

- P14, L25. With 4 hidden layers with 1,024 nodes each, your NN architecture will have millions of weights, a number that – if I am correctly interpreting the captions of Tables 1 and 2 – is about 1-2 orders of magnitude larger than the amount of data you used for training. This is often not considered good design practice in terms of overfitting avoidance. Are there any compelling reasons why you have chosen this architecture? Have you compared it to other architectural choices? Later on you say that this was done to handle the increase in the input layer dimensionality, but I do not see any obvious reason why an increase in the input layer dimension should require a similar increase in the hidden layers, especially because the main difference between your NN and the previous NN you are referring to is that you are no longer doing PCA, so your input vector has about the same information content as before.

- P14, L26. By "trained" do you mean "presented to the NN"? It doesn't make much sense to say that "a batch of samples is trained".

- P15, L12. Does the added noise variance reflect what is expected for RSP? Was the noise regenerated at each training cycle?

- P15, L22-23. The results for v\_e look a bit concerning. Di Noia et al. (2019) report good results for synthetic v\_e retrievals from POLDER (RMSE  $\sim$ 0.04), and RSP should be even better suited than POLDER for retrieving effective variance, as it can sample the rainbow region at more viewing angles and more wavelengths. Are you sure your underperformance is not caused by some design choice? Is the rainbow always sampled in the evaluation dataset?

- P16, L3-4. In what sense "the behavior of the initial output layer"? Do you mean the NN retrieval or what?

- P16, L20, sentence "in the framework of NN it is difficult to diagnose the source of this error". Please express more precisely what you mean. Which methods would you use
to diagnose it if you were using a different technique and why can't the same methods be used in the framework of NNs? Is the forward model used in the PP retrievals the same as the one you used to train the NN? If you pass your NN retrievals back to the forward model you used to train the NN, how does the synthetic measurement compare with the observation? And if you pass the PP retrievals to the same forward model? Furthermore, what happens if you pass the PP retrievals to the forward model and try to invert the output with the NN? If, instead, the forward model is different from the model used to create the PP database, and you feed your NN retrievals back to both, how do the simulated measurements compare?

- P18, L13-15, sentence "An evident feature ... below each PDF". I also see some spikes in the NJK histograms. Do they also correspond to LUT points? Going back to the NN histogram, could it be that your NN model is too flexible and does not generalize well outside the discrete grid points you used for training? After all, your NN is remarkably large (4 hidden nodes with 1024 neurons each sounds really huge to me). This again relates to my question how did you come to the conclusion that this NN architecture was suitable for your task. I would also recommend trying to sample the training data from a continuous distribution, but I see you already mention something like that in the conclusions.

- P20, L15. I guess "the behavior of" should not be there.

- P21, L4-5. You attribute the poor correlation in r\_e retrievals to the absence of SWIR data, but doesn't polarization give you sensitivity to r\_e even if you don't have SWIR, at least in the rainbow angles? Doesn't that mean that for your NN it is still difficult to find a good balance between using radiance and polarization in the r\_e retrieval? This goes back to the question of whether it would be better to try to infer r\_e with a NN that only uses polarization.

- P22, L1, about the flattening of the tau scatter plot for retrievals using SWIR. How does SWIR reflectance behave as a function of tau? Does it saturate for a smaller COT
than at shorter wavelengths? If so, this may explain why using SWIR is detrimental in your case. Later in the paper you say that SWIR channels are also more affected by 3D radiative effects, so is this maybe the reason?

- P25, L12, I guess there should be an "of" between "impact" and "atmospheric absorption"

- P28, L3. I guess "While" should be removed.

- P28, L17, statement "it lacks a clearly traceable relationship between observations and retrievals". I agree with you if you are establishing a comparison between NNs and simple retrievals such as Nakajima-King, where the relationship between observations and retrievals is so simple that it can be even represented visually. However, unless you can give a precise definition of "physical traceability", I do not totally agree if you are also comparing NNs to more complicated fitting schemes such as optimal estimation and Phillips-Tikhonov (as you seem to do two lines below), as the convergence of such schemes is also affected by a number of subjective choices (e.g., choice of covariance of regularization matrix, choice of the regularization parameter, choice of the first guess, choice of boundaries for the parameter values, and I am probably forgetting something), in an often unpredictable way. If you are referring to the fact that OE or PT retrievals optimize a clearly defined cost function involving the observation and the state vector, you should note that with an increasing number of training data the NN retrieval should asymptotically converge to the conditional expectation of the state vector given the measurements (Bishop, 1995) which, for example, for a linear Gaussian problem would coincide with the "traditional" OE solution (which, instead of the conditional expectation gives you the "conditional mode"). Thus also NNs have a "traceable" rationale behind them.

**REFERENCES**

Bishop, C. M. (1995), Neural Networks for Pattern Recognition, Oxford University Press.
Di Noia, A., et al. (2019), Retrieval of liquid water cloud properties from POLDER-3 measurements using a neural network ensemble approach, Atmos. Meas. Tech, 12, 1697-1716, doi: 10.5194/amt-12-1697-2019.

Nocedal, J., and Wright, S. J. (1999), Numerical Optimization, Springer.

**AMTD**

---

## Referee Comment (RC2) · Anonymous Referee #2 · 3 Dec 2019

This study introduced a neural network method to retrieve cloud optical thickness and effective radius efficiently from RSP especially when aerosol layer lies above clouds. To improve the unreasonable importance of input vector (total reflectance and polarized reflectance) suggested by PCA in training network, this study adjusted the weighing of total reflectance and polarized reflectance based on their uncertainty to assure the constrain of uncertainties. The application and test of the algorithm show good agreements with traditional LUT cloud retrievals for optically thick clouds but not very well suitable for thin, inhomogeneous or broken clouds. The paper is easy to read and well organized.

1. In section 2.3, the author introduced different characteristics of NJK and PP methods, given the high angular and spatial resolution of RSP measurements, I am not convinced that adding NJK method in the NN training might improve the accuracy. Even using more measurements generally lead to more reliable estimates about the unknow parameters, the measurement uncertainties between the total reflectance and DOLP is too large, which means introducing the total reflectance can also lead to uncertainty. 2. Regarding to the pre-processing of the input total reflectance and DOLP, the authors modified the calculation method of the inputs for the NN, using the measurement uncertainty to replace the standard deviation. The concern to assign different weighing for the total reflectance and DOLP is reasonable. However, I feel confused about the sentence "We found that the range of our standardized training set values for DoLP is roughly four time larger than that of RI. This means that, relative to measurement uncertainty, DoLP is approximately that much more sensitive to the parameters we vary in our training set than RI." My question is how to understand the four times difference for the two kinds of measurements. I guess the ratio can be further changed by adjusting the way to calculate the inputs (for example, further increase the weighing for DOLP measurements in equation 11 by introducing another factor). 3. Another thing is how to understand the comparison between PP, NKJ and NN results, which one is the truth? If the PP result is considered as truth, again the question is maybe only using DOLP to train the NN can get better results.

---

## Referee Comment (RC3) · Anonymous Referee #3 · 14 Jan 2020

The paper by Miller et al. developed a neutral network (NN) approach for estimating droplet size and cloud optical depth from a combined set of radiometric and polarimetric datasets that RSP acquired during ORACLES. Proper weighting is performed by accounting for uncertainties with total and polarized radiance measurements. To correct the retrieval bias with effective radius (as compared to standard polarimetric cloud retrieval), the algorithm applied a correction. On such a basis, the NN and standard parametric polarimetric (PP) cloud retrievals of RSP ORACLES 2016 data give consistent results, e.g. R = 0.756 and RMSE =  $1.74\mu$ m for droplet effective radius and R = 0.950 and RMSE = 1.82 for cloud optical thickness.

I read paper with much interests and have the following comments for the authors to consider:

1. The Nakajima & King approach uses the radiances in two bands for deriving an effective droplet size while polarimetric retrieval uses the angular distribution of cloudbow polarization to estimate droplet size. These two approaches are a) subjected to different error sources (e.g. 3D for NK approach and cloud-top region only for polarimetric retrieval) and b) may carry information about the cloud droplet size for different regions. By combining two different types of datasets of retrieval, it is possible that error sources couple with each other and it becomes harder to disentangle their impacts on the retrieval products. In this sense, the authors need to be more clear about the essence of performing a combined retrieval.

2. The difference of retrievals in Figure 6 and 7 stems from the correction of NN retrieval of effective radius using both radiance and polarization using PP based retrieval using polarization only. This means PP retrieval of effective radius is still used as more accurate and standard data. Then would it be more sensible to directly derive effective radius from PP method and apply the combined dataset retrieval to just get cloud optical depth ?

3. When weighting the total and polarized radiance in NN retrieval, were the modeling errors included ? As the authors pointed out, radiances more subjected to plane-parallel modeling errors (while polarization is less subjected). How do the authors account for such an effect at the weighting step ?

4. As the authors pointed out, the above-cloud aerosols are expected to impact the NN retrieval (especially the cloud optical depth part). Could some immediate work be done by performing a numerical test to assess its impact? For example, the authors can still use the plane-parallel model to generate radiance (but with the addition of absorbing aerosols above cloud). Then run the NN retrieval that excludes aerosols. With this extra work, it would be very helpful to track the aerosol induced errors and make their
analysis in the current work more robust.

5. One of the advantages of polarimetric retrieval is to determine the effective variance (veff) of cloud-top droplet size distribution, which means the information are in the polarimetric measurement. But on Page 5, the authors states "... an indication that this network cannot adequately retrieve veff"? I wonder if the NN algorithm somehow removes the information originally residing in the measurements.

Some editorial changes:

Abstract: the authors state that "This approach could be particularly advantageous for more complicated atmospheric retrievals such as when an aerosol layer lies above clouds like in ORACLES". But the above-cloud aerosol effect was not accounted for in NN retrievals presented in the paper.

Figure 6. Unit (microns) needed to be added to RMSE of effective radius in the legend.

P. 15: "...after the RMSE in the ve(ff) evaluation after training is enough to span the possible state space —an indication that this network cannot adequately retrieve ve(ff)". Here should the wording "enough" be "not enough"?

P. 28: Please add reference to the statement - "This is unlike the other RSP retrievals, which typically make use of a limited wavelengths and either polarized or total reflectance observations."

AMTD

---

## Author Comment (AC1) · 11 Feb 2020

Author Response: AMT-2019-327

We would like to thank the reviewers for their thoughtful comments. In response to the reviewer's suggestions we will make several important clarifications in the manuscript text. In addition to this, we will update the manuscript to include links to the public dataset.

Specific Comments:

1) We will revise this sentence to read as follows: "In this study we developed a neural

network (NN) that can be used to retrieve cloud microphysical properties from multiangular and multi-spectral polarimetric remote sensing observations."

2) P3, L33-34: Agreed, and this is a very good point. Our intention was for this sentence to indicate that the NN does not impose a specific analytical interpretation of the training data. We will revise this sentence as follows: "... in a manner that is independent of any imposed parameterized relationship between geophysical variables and the observations ..."

3) P7, L29. There is still some limited information about the cloud droplet size in the angular and spectral dependence of the total reflectance, but the SWIR bands provide the greatest sensitivity to droplet size information. The primary reason we wanted to develop a NN that mixed total and polarized reflectance information is to achieve a future objective (not the focus of this paper) of performing simultaneous aerosol and cloud retrievals using a similar framework.

4) P8, L9. In the revision we will modify to: "The HSRL-2 screening criteria were removed..."

5) P12, L10-11. The cloud top height was originally arbitrarily fixed to limit the size of the training dataset. Also, the cloud top to observation altitude (flight altitude) separation is actually the more important difference impacting the cloud retrievals. By fixing the cloud top height, which is relatively consistent throughout the campaign, the difference in cloud top and observation altitude is modeled.

6) P13, Table 1. As in the previous work (Segal-Rozenhaimer et al (2018)) we assumed a black ocean, and therefore no dependence upon wind speed or ocean color. Considering that the smallest cloud optical depth in our training set was 2.5, changes from this assumption would have a minimal impact on the observed radiances, and an even smaller impact on the polarimetric observations. Furthermore, the expression of the cloud bow in the polarimetric signal is quite different than that of a glint (peaked in the reflected sun direction) or ocean color (likely isotropic) reflection. For these reaAMTD
sons, we concluded that training for those parameters as well would not be worth the computation effort. Segal-Rozenhaimer et al (2018) described these approximations in section 4.1, and it wouldn't hurt for us to summarize in this paper. So, we added the following text to the end of section 3.1:

"For reasons of computational efficiency, the training set simplifies some aspects of nature. Compared to the predecessor paper (Segal-Rozenhaimer et al., (2018), we used a larger set of geometries and wider range of parameter values, but many of the same approximations. For example, this training set assumes plane parallel radiative transfer (neglecting 3D effects) and a 'black' ocean surface with no reflections due to sun glint or ocean color. The former is beyond our computing resources and desired level of parameterization, while the former is expected to be heavily attenuated by the cloud."

7) P13, L7, "principle" was amended to "principal". Thank you for catching this error.

8) P14, L7-9. We have attempted to parse this more carefully in the revised manuscript.

9) P14, L17-20. Our approach to standardization was performed as a two-step process: the first is the standardization based on the uncertainty and the scale of the different inputs, and the second is the linear scaling step, to put the inputs in a range between about -1 and 1. We chose to perform these steps separately since we know the uncertainty model for our inputs. Indeed, although the common practice is to scale/normalize all inputs similarly (scaling all covariances to the same values), in the case where some inputs have different contribution (in our case, the reflectance data has larger uncertainty so we wanted it to be less "visible"), it is advisable to scale the inputs differently (LeCun et al., Efficient BackProp, 1998). One might argue that we could have done this in one step but we felt that this approach has a better physical basis (i.e. using the uncertainty model to rescale the inputs and then the "standard" scaling to assist in the convergence of the system, as in the common practice.

10) P14, Section 3.3. While we agree, the evaluation of those retrievals has been

**AMTD**
performed before on other similar simulated datasets (e.g., Alexandrov et al. 2012a,b and Miller et al. 2018), and the behavior of our training set can likely be approximated from those previous studies – which both used the same forward radiative transfer model.

11) P14, L25. We have compared numerous architectural choices in the process of arriving at the one used in this research (including the architectures we experimented with throughout the development of our previous paper, Segal Rozenhaimer et al. (2018) as you mention). We tested architectures with many fewer hidden nodes (between 40 and 1024), including varying the number of hidden layers and got best results with the architecture we are currently using. Indeed, although PCA theoretically represent the majority of the variance, it isn't always a one-to-one physical representation, which seemed to be important in our case (PCA reff results were less generalized than the current one), since the RSP has a large number of viewing zenith angles, and their full representation seemed to add more information. The number of hidden layers and nodes is often a trial-and-error parameter for each problem. Hence, we performed cross-validation on the various architectures tested and got the best training and test results for the selected architecture. We tested drop-offs as well, but did not get better results, or greater generalizability.

12) P14, L26. We have revised this to say: "This network is instead trained using a mini-batch method, where a batch of samples (128) is presented to the network and the weights on each of the hidden layers are only updated after each batch has been processed.

13) P15, L12. Noise is generated each training cycle and is representative of radiometric uncertainty. However it is not based on the instrument uncertainty model used in our uncertainty standardization process described earlier in the paper.

14) P15, L22-23. Even in our previous network we were not getting reliable results for ve (compared to PP retrieval of ve). The most likely explanation is that when the NN is
simultaneously optimized for tau, re, and ve the retrieval the behavior of the ve retrieval suffers the most because there is simply less sensitivity to this parameter than to re or tau. Also, the PP retrieval can often struggle to retrieve ve as well (refer to figure 4 of Miller et al. 2018). I think a NN result trained for only a single variable retrieval, such as in the work of Di Noia et al. (2019), would perform a more accurate ve retrieval. However, because the objective of our network is to eventually disentangle the coupled retrieval of aerosol and cloud properties, we have attempted to avoid single variable retrievals.

As far as angular sampling is concerned, we are indirectly using the same criteria as the PP retrieval – because we have conditioned our retrieval on the comparison to an existing PP retrieval. This means that we are always seeing rainbow angles in the results shown here. We used to have a stricter angular sampling condition (only near the principal scattering plane), but we have relaxed that requirement and increased the size of the training set in this study. This relaxation of angular requirements increases the amount of observations we can retrieve but makes the retrieval problem more complicated.

15) P16, L3-4. Sorry, this is a confusing attempt to distinguish the output of the NN and the linearly adjusted output of the NN. We have attempted to clarify the language in the revision.

16) P16, L20. We have reframed this statement. It was originally intended to convey the sense that machine learning can sometimes feel like a black box. However, this is something that we've come to learn isn't entirely the case and there have been significant recent advancements in the field of "explainable AI" (e.g., McGovern et al. 2019).

Your recommendation regarding the relationship between the forward model and the NN retrievals is an interesting one. We are indeed using the same forward model in both cases. However we have not looked at this sort of inverse problem analysis
because the greatest source of error we expect from this retrieval comes in the from assumption biases (3-D radiative effects, and other un-modeled features in the data) and less from forward modeling biases themselves.

17) P18, L13-15, in reference to sentence "An evident feature . . . below each PDF". I assume you were referring to the re histograms here. It's possible the "spikes" in the NJK histogram are actual features of the distribution of microphysical properties because I also see similar features in the RFT and PP (lower bin resolution) histograms.

We agree, the current network implementation poorly generalizes from the limited LUT grid. We have modified this approach for our future studies to randomly sample a continuous distributions of physical parameters, but we have not made this change for this particular study.

18) P20, L15. "I guess "the behavior of" should not be there." Thank you, we have corrected this in the revision.

19) P21, L4-5. This was perplexing, but I think the explanations for it are two-fold. First, it could be that mixing total and polarized reflectance information is still not working well in our network – the polarized reflectance information is still not weighted strongly enough. Second, we think that this could be a result of the network lacking an explicit understanding of the joint relationship between the spectral and angular dependence. Without this shared dependence the network is treating measurements at different wavelengths and angles as independent pieces of information without necessarily relating them to one another. To this end, our future work on this topic will likely involve a convolutional neural network approach, where the multi-angle and multi-spectral dependence of the training data can be expressed as an image – explicitly providing the neural network with context for the joint dependence of the dataset on spectral and angular information.

As mentioned previously, we were attempting to avoid a total or polarized reflectance only retrieval because our final objective (aerosols above clouds) will require us to AMTD
disentangle information from both total and polarized reflectance. Perhaps for the microphysical portion of this problem though, it may make sense in the future to perform the re/ve retrieval on polarimetric only measurements.

20) P22, L1. SWIR bands also increase in brightness with increasing optical thickness, but they are generally less sensitive to this change than the VNIR bands. They also saturate at lower optical thicknesses. Later in the paper, I was attempting to refer to the other wavelength bands that are not normally used in the bispectral retrieval (0.410 - 0.555  $\mu$ m). We will attempt to clarify the sentence that says that in the revised paper to avoid future confusion.

21) P25, L12, "I guess there should be an "of" between "impact" and "atmospheric absorption" " Thanks, this was corrected in our revision.

22) P28, L3. I guess "While" should be removed. We have removed while and replaced it with "Whereas"

23) P28, L17, statement "it lacks a clearly traceable relationship between observations and retrievals". These are good points. We will revise the section to be less aggressive about our usage of physical traceability when discussing the Neural Network approach. We will attempt to make a subtler point in this section.

REFERENCES Reviewer Bishop, C. M. (1995), Neural Networks for Pattern Recognition, Oxford University Press.

Di Noia, A., et al. (2019), Retrieval of liquid water cloud properties from POLDER-3 measurements using a neural network ensemble approach, Atmos. Meas. Tech, 12, 1697-1716, doi: 10.5194/amt-12-1697-2019. Nocedal, J., and Wright, S. J. (1999), Numerical Optimization, Springer.

**REFERENCES** Author**

Alexandrov, M. D., Cairns, B., Emde, C., Ackerman, A. S., and van Diedenhoven, B.: Accuracy assessments of cloud droplet size retrievals 10 from polarized reflectance
measurements by the research scanning polarimeter , Remote Sensing of Environment, 125, 92–111, 2012a.

Alexandrov, M. D., Cairns, B., and Mishchenko, M. I.: Rainbow Fourier transform, Journal of Quantitative Spectroscopy and Radiative Transfer, 113, 2521–2535, 2012b.

Miller, D. J., Zhang, Z., Platnick, S., Ackerman, A. S., Werner, F., Cornet, C., and Knobelspiesse, K.: Comparisons of bispectral and polarimetric retrievals of marine boundary layer cloud microphysics: case studies using a LES–satellite retrieval simulator, Atmospheric Measurement Techniques, 11, 3689–3715, 2018.

McGovern, Amy and Lagerquist, Ryan and John Gagne, David and Jergensen, G. Eli and Elmore, Kimberly L. and Homeyer, Cameron R. and Smith, Travis, Making the Black Box More Transparent: Understanding the Physical Implications of Machine Learning, Bulletin of the American Meteorological Society, 100, 11, 2175-2199, 2019, 10.1175/BAMS-D-18-0195.1

Segal-Rozenhaimer, Michal, D.J., Miller, K. Knobelspiesse, J. Redemann, B. Cairns, M.D. Alexandrov, Development of neural network retrievals of liquid cloud properties from multi-angle polarimetric observations, Journal of Quantitative Spectroscopy and Radiative Transfer, 220, 2018, 39-51, DOI:10.1016/j.jqsrt.2018.08.030.

AMTD

---

## Author Comment (AC2) · 11 Feb 2020

Author Response: AMT-2019-327

We would like to thank the reviewers for their thoughtful comments. In response to the reviewer's suggestions we will make several important clarifications in the paper text. In addition to this, we will update the paper to include links to the public dataset.

Specific Comments:

1) The NJK method is not implicitly used in the training of the network. The network is simply trained based on the total reflectance and polarized reflectances in a number

of spectral bands and scattering geometries. However, we do agree that the greater uncertainty in the total reflectance has been one of our primary difficulties in this research. It is also important to note that without total reflectances there is no method for obtaining an accurate estimate cloud optical thickness. Refer to my answer to your third question for some expanded discussion as to why we did not want to separate total and polarized reflectances in the implementation of this network.

2) We understand the source of the confusion, it is a rather tortured explanation. We could of course modify the input vector in whatever manner we choose, but the idea here is to modify the relative importance of each measurement by the uncertainty. It is also important to understand that we are not simply replacing standardization in the normalization process, we are just introducing a pre-processing step before submitting data to the network. Batch renormalization occurs throughout the network, as indicated in our network diagram in figure 3.

We have clarified this statement in our revision: "After this standardization relative to instrument uncertainty the range of variability in the DoLP training set input is approximately four times greater than the range of variability in the RI. This variability difference is a result of the better relative measurement uncertainty of DoLP compared to RI. As a result the network should initially place a greater weight on changes in DoLP than on changes in RI."

3) It is a difficult question, as both of the baseline retrievals exhibit their own uncertainties and dependence on observational conditions. However, as we mentioned in the paper in the last two paragraphs of section 2.3 (page 10-11), our previous work in Miller et al., 2018 showed that a synthetic comparison of the bispectral (NJK) and polarimetric (PP) retrievals revealed that at high spatial resolutions obtained by airborne instruments like RSP the retrievals should be nearly identical for optically thick clouds. This indicates that the information content (with regard to re and tau) contained in these two measurements is relatively similar. Additionally, we examined this in the background section by examining the comparison of NJK and PP for both field campaigns in this study and found similar results for optically thick clouds (shown in figure 2).

While it is possible to use only one type of measurement to train a network retrieval on, but the two different type measurements (total and polarized reflectances or DoLP) contain additional information content about aerosols. One of the main reasons we wanted to mix them in our retrieval stems from our eventual final objective of this research program – a NN estimate of aerosol above cloud and cloud properties. The existing robust methods of inferring above cloud aerosol properties require both total reflectance and DoLP to tease out the different signals of the above cloud aerosol and the cloud itself. For the purposes of our future exploration of the NN approach we needed to understand how mixing these two types of information into one NN might influence our approach.

---

## Author Comment (AC3) · 11 Feb 2020

Author Response: AMT-2019-327

We would like to thank the reviewers for their thoughtful comments. In response to the reviewer's suggestions we will make several important clarifications in the paper text. In addition to this, we will update the paper to include links to the public dataset.

1. There are two aspects here that require response separately: First, the two retrieval approaches have different error sources and different information; and Second, that combining the two types of datasets is sensitive to error sources in each, making it

difficult to disentangle impacts.

a) Two Approaches: The reviewer is correct to note that the different retrievals come along with different sensitivities to different sources of error and uncertainty. In addition to this they are also sensitive to different effective sensing depths within the cloud, though this statement is largely dependent on that bands selected for the bispectral retrieval. For strongly absorbing bands 3.75, the differences in effective sensing depth are less severe. However, those statements being acknowledged, the work of Miller, et al. (2018), demonstrated that for the differences between the two retrievals (at high spatial resolution) tend to be minimal for unbroken clouds that are spatially homogeneous, optically thick, and have microphysically narrow cloud droplet size distributions. These statements also happen to be representative of the cloud regime marine boundary layer stratocumulus we observed throughout the ORACLES field campaign. In the paper we also discuss these different sources of conflict between the two retrievals and demonstrate in our analysis that they are still relatively similar to one another in figure 2.

b) Disentangling Impacts of biases: On this front, I think that our approach runs into difficulties as the reviewer has pointed out. It is certainly already difficult (though not impossible) to disentangle the sensitivities of machine learning approaches to the properties and sensitivity to error in input state vector. Some studies regarding the development of more understandable and physically traceable machine learning approaches have been more recently undertaken. McGovern, et al. 2019 is a good example of the kind of work being performed to address understandability of machine learning. However, this field has been growing extremely rapidly, and as a consequence it expanded a lot the paper well after the paper was already finished.

Despite being a possible source of difficulty, mixing the information content of the two radiometric techniques helps us to achieve the stated objectives of this work – namely to provide a quick a priori input to retrieval methods that already must mix radiometric and polarimetric information to address complicated remote sensing problems (i.e.,

simultaneous retrieval of aerosols above clouds). In ORACLES, smoke aerosols are commonly observed over clouds, and the intended objective of studying this dataset is to improve future simultaneous retrievals.

2. This is not something that we considered, but the proposed approach is certainly sensible. The reviewer is correct in noting that the most obvious difference between the corrected results and the uncorrected results is the linear offset of the effective radius retrieval. However, as we explored a lot of different sorts of network architectures we found that this one demonstrated the most significant improvement in the re RMSE from our previous study, Segal-Rozenhaimer, et al. 2018. In the previous study we were unable to get very sensible re retrievals and concluded that the approach probably wasn't very useful for re retrievals. In contrast to this conclusion, when we found in this work that there was only systematic high bias in the NN retrieval it was quite a positive revelation. Additionally, as we later indicate in the discussion section, it is possible to improve the neural network without this linear correction approach – but instead by applying atmospheric correction for gaseous absorption above the cloud prior to input to the neural network. We saw greater improvement in the initial output of the network after performing this correction, especially for ORACLES 2017 (and later 2018, not included in the paper), when the separation between cloud top and the airborne platform (NASA P3) was far more variable.

3. The weighting is strictly based on the RSP instrument uncertainty model and not on anything else. The impact of the forward model assumed geometry plane-parallel infinite slab (or rather 1-D radiative transfer), is not accounted for and is therefore treated as an assumption error. The reason this is the case is because there is not yet, to our knowledge, a generally accepted method for dealing with 3-D radiative effects in a single-pixel retrieval. Some techniques exist using iterative 3-D radiative transfer modeling, but require providing a whole cloud domain and performing all retrievals at the same time.

4. The explicit inclusion of above cloud aerosols in this research is part of our future

plans. However, because this was intended as a research algorithm to demonstrate our ability to perform and validate a new Neural Network cloud retrieval against existing algorithms – which also exhibit this source of error – tracking down this error source is outside the scope of this analysis. However, given that our comparison to the other RSP COT retrievals is rather good, it indicates that it is likely that the results of such a study would be very similar to those indicated in prior work by Meyer, et al. (2013). In that work, they showed that the bispectral retrieval of cloud optical thickness is biased low due to the presence of an absorbing above-cloud aerosol, but that there is minimal impact on the bispectral re retrieval. This indicates that the same impact would be present in the polarimetric cloud optical thickness retrieval.

To address this in the manuscript, we will revise the paper to more explicitly discuss the results of Meyer, et al. (2013) in the context of our work.

5. This was kind of puzzling, but we hypothesize that it is because the magnitude of the primary cloud bow is sensitive to both COT and ve when the COT is less than 3. Therefore this particular observation geometry has a non-linear dependence on two different retrieval variables. As a result of this joint dependency, the network manipulates the observed information in favor of reducing the uncertainty in either the COT or the ve and here it appears to favor the COT determination.

Some editorial changes: * As mentioned within the text of the manuscript, this study is intended as a required first-step toward the objective of a future machine learning approach for the simultaneous joint-retrieval of aerosol and cloud properties for ACA scenes. We wanted to demonstrate that we could use Machine Learning to address the cloud-only approach before we made the state space more complicated.

* Figure 6. Unit (microns) needed to be added to RMSE of effective radius in the legend. This figure will be modified before final publication to include units on the RMSE

*P. 15: ". . .after the RMSE in the ve(ff) evaluation after training is enough to span

the possible state space an indication that this network cannot adequately retrieve ЁĞ ve(ff)" This was correct as written, but we have clarified this sentence to make it more clear overall. We were attempting to say that because the RMSE of veff is larger than the range of the training set a retrieval of veff in this framework would have extremely low skill. Ideally the RMSE would be small relative to the range of the training grid, as is the case for effective radius and optical thickness.

"The results for tau and re are quite promising, but the results for ve are concerning. The RMSE for ve is larger than the range of training parameter space listed in Tables 1 and 2, an indication of both high uncertainty and very low precision. As a consequence, we do not consider this network capable of inferring adequate information about ve."

P. 28: "Please add reference to the statement - "This is unlike the other RSP retrievals, which typically make use of a limited wavelengths and either polarized or total reflectance observations."" References to both of the RSP retrieval papers have been added to the revised manuscript. These are Alexandrov et al. 2012a and Alexandrov et al. 2012b respectively (and have been cited throughout the paper). This also contains a typo and has been corrected to read:

"This is unlike the other RSP retrievals, which currently only make use of limited spectral information in each individual retrieval; both polarimetric retrievals use a single band (Alexandrov et al. 2012a, 2012b), and the bispectral retrieval uses a pair of bands."

REFERENCES Author

McGovern, Amy and Lagerquist, Ryan and John Gagne, David and Jergensen, G. Eli and Elmore, Kimberly L. and Homeyer, Cameron R. and Smith, Travis, Making the Black Box More Transparent: Understanding the Physical Implications of Machine Learning, Bulletin of the American Meteorological Society, 100, 11, 2175-2199, 2019, 10.1175/BAMS-D-18-0195.1

[Figure]

Meyer, K., S. Platnick, L. Oreopoulos, and D. Lee (2013), Estimating the direct radiative effect of absorbing aerosols overlying marine boundary layer clouds in the southeast Atlantic using MODIS and CALIOP,J. Geophys. Res.Atmos.,118, 4801–4815, doi:10.1002/jgrd.50449.

Segal-Rozenhaimer, Michal, D.J., Miller, K. Knobelspiesse, J. Redemann, B. Cairns, M.D. Alexandrov, Development of neural network retrievals of liquid cloud properties from multi-angle polarimetric observations, Journal of Quantitative Spectroscopy and Radiative Transfer, 220, 2018, 39-51, DOI:10.1016/j.jqsrt.2018.08.030.

---

## Author Response (AR2)

**Author Response: AMT-2019-327**

We would like to thank the reviewer again for their thoughtful comments. In response to the reviewer's suggestions we have added more clarifications in the paper text.

*Reviewers comments formatted in italics*
Authors responses formatted in plain text

**Anonymous Referee #1**

*The Authors have made an effort to revise their manuscript according to the suggestions from reviewers. Personally, I feel that the paper would benefit from another round of revisions, as I still have a few questions about the Authors' response to my comments. I will mark this paper for minor revisions, as my perception of this work is generally positive, but still I advise the Authors to pay attention to a few points that – in my opinion – would still benefit from some additional revisions. In general, I feel that some parts of the paper may give the readers the impression that some issues still open in the retrievals are weaknesses inherent to "the neural network approach" in general, when in reality they may well be caused by some particular design choices. Once again, I am referring to: use of relatively few training data and a very large network, some assumptions in the training set that may limit its comprehensiveness. This is a message that I would like to avoid. After the discussion is made a bit more balanced, I think this work can be published, because the results shown in this paper are certainly interesting to the scientific community.*

**Replies to Specific Points**

- *5) Can you add this consideration to the paper, and possibly add some references to back this statement? Furthermore, in your ER-2 NN you also use a fixed flying altitude (20 km). Therefore, unless I am missing something, the impact of cloud height variations (or cloud-aircraft altitude difference) on the top-of-atmosphere signal is not accounted for at all, and I think that this limitation needs more emphasis in the text. In your second NN you vary the flight altitude between 5 and 7 km and keep the cloud height fixed at 1 km. I guess that low-level clouds typically have altitudes of, let's say, 1 to 5 km (also depending of what you consider to be a "low-level" cloud). Since the atmospheric density decays exponentially with height, qualitatively I would expect a 1 km change in cloud height between 3 and 5 km to impact the shielding of Rayleigh scattering underneath more than changing the flight altitude by 1 km. Of course mine are just qualitative considerations, but I wonder if there are some references supporting your line of reasoning. On top of that, I still think that in absolute terms your training set is rather small compared to other existing studies, which use training sets containing millions to even tens of millions of data (Kox et al., 2014, Strandgren et al., 2017, Di Noia et al., 2019). I would suggest to at least emphasize in your text that this may be a*

*limitation in your design setup.*

*References:*
*Kox et al. (2014), "Retrieval of cirrus cloud optical thickness and top altitude from geostationary remote sensing", Atmos. Meas. Tech., 7, 3233–3246, doi: 10.5194/amt-7-3233-2014*

*Strandgren et al. (2017), "Cirrus cloud retrieval with MSG/SEVIRI using artificial neural networks", Atmos. Meas. Tech., 10, 3547–3573, doi: 10.5194/amt-10-3547-2017*

- o You are correct regarding the idea of Rayleigh shielding by low-level clouds, that would indeed not be included in our approach for generating the training dataset for ORACLES 2017/2018. We have included in the manuscript a more explicit discussion of the limitations to our approach of incorporating varying cloud-to-platform separation in the training dataset.

- o As far as training dataset size is concerned, we have included a statement in our summary to emphasize that we might be using too-little training data and cited the indicated research in your response. In future work, we obviously will be considering these lessons regarding training data sampling and training data size. We have also revised the tables describing the training datasets to explicitly indicate the total scale of the dataset provided as input to the NN. Previously we only indicated the number of data labels in the NN training dataset (feature vectors) and not the full scale of the features in each vector. Hopefully this places the size of the input vector front and center before the discussion of the size of the NN – even if there are fewer relative training cases than the reviewer has indicated might be warranted for this approach.

- *9) LeCun et al. (1998) indeed seem to suggest that this approach may work, even though they do not explain why in mathematical terms. I am maybe being overscrupulous here, but I would still suggest to test whether your approach worked by looking at the statistics for the derivatives of your NN output with respect to each input, divided by the input standard deviation. This test should be simple to implement, and will tell you for sure whether or not your NN is really less sensitive to reflectance than it is to DoLP as you wish to achieve.*
  - o

- *11) I agree that the choice of the number of hidden layers is a trial-and-error procedure. However, there are certainly limits (also theoretical - see, e.g., Baum and Haussler, 1989, Haykin, 1999) to the ratio between the number of training samples and the number of free parameters in order to get a reasonable generalization. Usually one wants to have at least an order of magnitude more training data than free parameters, whereas for your NN the opposite is the case. Now: I guess you determined the number*

*of neurons on a subset of synthetic data not used during the training phase, and you may have found that your large NN architecture is the one that achieved the lowest RMS error. Is this the case? However, the real question is: how confident are you that your choice - determined through this procedure - is robust enough to be also valid for application to real data, which probably follow a statistical distribution that is very different to that of your training set? This question becomes especially important considering that your training set appears to contain a number of simplifications that may limit its comprehensiveness.*

- o If we understand your point correctly, we have approximately 1.7x the number of nodes in each individual training feature vector presented to the network. With many thousands of labeled training vectors (depending on 2016/2017 network). While it is not significantly more, it is still appreciably larger.

- o You are correct regarding your description of how we arrived at the large NN architecture – it was the architecture we landed on, through trial and error, which provided the lowest RMS error relative to other versions. It may also be important to note that many of these decisions were based on the strong positive results we saw in the 2016 dataset before we knew if we would have an architecture that would work in future years.

- o We believe that your question of robustness for application to real data is at the core of any neural network study – though we aren't sure how our study in particular is to solve a problem that basically any NN study would have difficulty addressing when the training and observation data come from different sources with different uncertainties (observations) and assumptions (training data). This is especially true when fully replicating the behavior of the observation dataset simply isn't feasible (e.g., 3D Radiative effects and cloud inhomogeneity). Those last two examples are likely bigger sticking points preventing the observational dataset from fully resembling the training data than even the lack of atmospheric correction or fixed cloud top height.

- *14) Di Noia et al. (2019) do not perform a single-variable retrieval of effective variance. They retrieve effective variance, effective radius and cloud height together. Separate retrievals are only reported for cloud optical thickness, as COT cannot be retrieved from polarized radiance alone. I still suggest to acknowledge in your text that your effective variance retrievals look less accurate than results already published in literature, both*

*NN (Di Noia et al., 2019) and non-NN (Shang et al., 2019) retrievals. Furthermore, could it be that your statistics for effective variance improve if you compute them on a subset of test data, e.g., only data in the principal plane?*

*Reference: Shang et al. (2019), "An improved algorithm of cloud droplet size distribution from POLDER polarized measurements", Remote Sens. Environ., 228, 61-74, doi: 10.1016/j.rse.2019.04.01*

- o Apologies, I did not intend to apply that the Di Noia paper was performing a ve-only retrieval. I was implicitly referring to the approach of separating out a variable to obtain a retrieval that does not dominate the feature-space variability of your input data (like COT in polarized reflectances). Admittedly though, the full separation of total and polarized reflectances might be the more relevant lesson from that paper.

- o We have added a statement to the paper, where we previously state that the NN is not capable of retrieving effective variance, that communicates this issue more xdirectly. In particular we refer to the Di Noia, 2019 paper. We also point out that we compared the ve(NN) retrieval of to RSP retrievals which are known to have good accuracy in both simulated and observational studies (Alexandrov et al. 2012a, Alexandrov et al., 2018).

- o We evaluated the $v_e$ retrieval output against several sampling filter criteria (for the same trained network) and did not see much improvement. We also explored different input vector combinations of radiometric variables – though in our previous work Segal-Rozenhaimer et al. (2018) we did find that if we provided both polarimetric variables as the NN input (Q and DOLP for example) the RMSE of the $v_e$ retrieval improved, but the behavior of the bias was still erratic and other retrievals ($r_e$/tau) were much worse. It may be important to separate the optimization of COT and these microphysical parameters if providing both total and polarized reflectances as inputs.

- *16) I have to admit that I do not see much reframing of this sentence in the text. I still see the sentence "in the framework of NN it is difficult to diagnose this error". However, you still do not explain WHY, in your opinion, this difficulty is typical of the NN framework as opposed to other retrieval schemes, considering that also NN retrievals can be fed back to radiative transfer models, derivatives of NN outputs with respect to inputs can be computed analytically, the response of the NN to specific inputs can be tested, etc. Therefore, I would still like to know what are the "error diagnosis techniques" that you can apply to other methods but not to a NN retrieval. Even if carrying out the analysis I recommended goes perhaps beyond the scope of your paper, I would still advise you to at least avoid generic statements such "in the NN framework it is difficult to diagnose this kind of error", unless you can explain what exactly is difficult and why.*

o We have removed this statement because we now believe that the source of the linear offset actually comes from the lack of atmospheric correction in our initial analysis. This statement was inserted into the paper before we discovered the found that atmospheric correction for the above-cloud gaseous absorption resulted in nearly unbiased NN retrievals. Instead we have modified the paragraph to read as follows:

" This linear bias was absent during our training set validation exercise in section 3.3, implying that this systematic offset is consequence of differences between training set and observational data. Despite this linear bias, the high correlations of these retrievals imply that the NN retrieval is otherwise generally performing correctly. In particular we expect that this bias is an expression of a difference between the assumptions built into the network training dataset that differ from the observation dataset. We will discuss the possible sources of these differences in section 5."

- *20) If you confirm that SWIR bands saturates at lower COTs than VNIR bands, then I would suggest mentioning this as a likely cause for the behaviour of your scatter plots, as attempting to invert a relationship that saturates typically gives this kind of effects.*

  o More discussion on the saturating COT behavior has been added to the paper. This is also common feature of the retrieval sensitivity in the bispectral approach – which mixes information content from bands that have very different sensitivities to optical depth. Also, a similar behavior exists for effective radius – but it is also dependent on scattering geometry as well (due to phase function dependence).

- *23) I do not perceive a big difference to the previous version of the paper. I still see a reasoning based on the concept of "clear traceability", which still looks somewhat vague to me. Let me be clear: it is not my intention to dismiss your point, but I would like to see a more focused discussion of what you actually mean by that.*
  *A possible hint for revising your discussion is that of conveying the concept that NN retrievals - as opposed to curve fitting, LUTs, or iterative (regularized) least squared retrievals - are not designed to obtain the best fit between simulations and observations for each observation. In my opinion, this would be a more valid point than generically saying that NNs are "less rigorous" or "less traceable" than other retrieval methods.*

  o Apologies, we have attempted to avoid the generalities discussed here and taken the advice of the reviewer. We've reframed the statement in the paper as follows:

  "The NN approach outlined here does come with some particular challenges that are unique relative to other approaches (e.g., LUT-based search, curve fitting, iterative least squares etc.) is that it is not designed to obtain a "best fit" between simulations and observations for *each* observation. Rather it is designed to obtain a "best fit" for a population of training data and later generalize that behavior to different observational input data. As a consequence, analyzing the behavior of

NN retrievals requires carefully developing an understanding of the training and input datasets and their potential differences. However, despite this potential difficulty, the NN retrieval was shown here to provide reasonable results, lending support to the idea that a NN retrieval could provide a quick first guess to other numerically rigorous retrievals (Di Noia et al., 2015)."

**MINOR REVISIONS**

- *P2, L30. Cite Werdell et al. (2019), "The Plankton, Aerosol, Cloud, ocean Ecosystem (PACE) mission: Status, science, advances". Bull. Am. Meteor. Soc., 100, 1775-1794, doi:10.1175/BAMS-D-18-0056.1*
  - Revised to include this citation, thanks for the catch.

- *P13, L18. I guess the second "the former" in the sentence should be replaced with "the latter"*
  - Revised to correct this error.

[revised manuscript text omitted]